

# A simplified parameterization of isoprene-epoxydiol-derived secondary organic aerosol (IEPOX-SOA) for global chemistry and climate models

Duseong S. Jo[1,2], Alma Hodzic[3,4], Louisa K. Emmons[3], Eloise A. Marais[5], Zhe Peng[1,2], Benjamin A. Nault[1,2], Weiwei Hu[1,2], Pedro Campuzano-Jost[1,2], and Jose L. Jimenez[1,2]

[1]Cooperative Institute for Research in Environmental Sciences (CIRES), University of Colorado, Boulder, CO, USA
[2]Department of Chemistry, University of Colorado, Boulder, CO, USA
[3]Atmospheric Chemistry Observations and Modeling Lab., National Center for Atmospheric Research, Boulder, CO, USA
[4]Laboratoire d'Aérologie, Université de Toulouse, CNRS, UPS, Toulouse, France
[5]Department of Physics and Astronomy, University of Leicester, Leicester, UK

*Correspondence to*: Jose L. Jimenez (jose.jimenez@colorado.edu)

**Abstract.** Secondary organic aerosol derived from isoprene epoxydiols (IEPOX-SOA) is thought to contribute the dominant fraction of total isoprene SOA, but the current volatility-based lumped SOA parameterizations are not appropriate to represent the reactive uptake of IEPOX onto acidified aerosols. A full explicit modelling of this chemistry is however computationally expensive owing to the many species and reactions tracked, which makes it difficult to include it in chemistry climate models for long-term studies. Here we present three simplified parameterizations for IEPOX-SOA simulation, based on an approximate analytical / fitting solution of the IEPOX-SOA yield and formation timescale. The yield and timescale can then be directly calculated using the global model fields of oxidants, NO, aerosol pH and other key properties, and dry deposition rates. The advantage of the proposed parameterizations is that they do not require the simulation of the intermediates while retaining the key physico-chemical dependencies. We have implemented the new parameterizations into the GEOS-Chem v11-02-rc chemical transport model, which has two empirical treatments for isoprene SOA (the volatility basis set (VBS) approach and a fixed 3% yield parameterization) and compared all of them to the case with detailed full chemistry. The best parameterization (PAR3) captures the global tropospheric burden of IEPOX-SOA and its spatio-temporal distribution ($R^2 = 0.93$) vs. those simulated by the full chemistry, while being more computationally efficient (~5 times faster), and accurately captures the response to changes on NOx and SO₂ emissions. On the other hand, the constant 3% yield that is now default in GEOS-Chem deviates strongly ($R^2 = 0.65$, 63% underestimation), as does the VBS ($R^2 = 0.45$, 78% underestimation), with neither parameterization capturing the response to emission changes. With the advent of new mass spectrometry instrumentation, many detailed SOA mechanisms are being developed, which will challenge global and especially climate models with their computational cost. The methods developed in this study can be applied to other SOA pathways, which can allow including accurate SOA simulations in climate and global modeling studies in the future.





# 1 Introduction

Secondary organic aerosols (SOA) are a major component of submicron particulate matter globally (Zhang et al., 2007; Jimenez et al., 2009), but are typically poorly predicted by global models (Tsigaridis et al., 2014). Isoprene is the most abundant non-methane volatile organic compound (VOC), whose global emission flux (~600 Tg yr$^{-1}$) is much larger than that of monoterpenes (~100 Tg yr$^{-1}$) (Sindelarova et al., 2014) and non-methane VOCs from anthropogenic sources (~130 Tg yr$^{-1}$)  (Lamarque et al., 2010). On

account of its global source strength, isoprene oxidation can contribute substantially to SOA in the atmosphere, even if its yield is small (Carlton et al., 2009). There are several isoprene oxidation products that can lead to SOA formation, including isoprene-derived epoxydiols (IEPOX) (Paulot et al., 2009), glyoxal and methyl glyoxal (Fu et al., 2008), gas-phase low volatility organic compounds (LVOC) produced from gas-phase oxidation of hydroxy hydroperoxides (ISOPOOH) (Krechmer et al., 2015; Liu

et al., 2016), and methacryloylperoxynitrate (MPAN) (Surratt et al., 2010). Gas-phase IEPOX, mainly formed from the photooxidation of isoprene under low NO conditions (Paulot et al., 2009), can efficiently partition onto aqueous acidic aerosols and produce SOA through aqueous-phase reactions (Paulot et al., 2009; Surratt et al., 2010; Gaston et al., 2014a; Zhang et al., 2018). SOA from IEPOX ("IEPOX-SOA") is considered at present as the dominant isoprene-derived SOA pathway (Marais et al., 2016; Carlton et

al., 2018; Mao et al., 2018), compared to a less efficient formation from glyoxal (Knote et al., 2014).

  Ground-based and aircraft field measurements have shown that IEPOX-SOA can contribute to total OA concentrations by as much as 36%, especially for forested regions under low NO across the globe (Hu et al., 2015). Several modeling studies have explicitly simulated IEPOX-SOA by considering detailed isoprene gas-phase chemistry and IEPOX uptake (Marais et al., 2016; Budisulistiorini et al., 2017;

Stadtler et al., 2017). Figure 1 shows the main chemical pathways of the IEPOX-SOA chemistry in (a) HO$_2$ and (b) NO dominant conditions simulated by GEOS-Chem. The fate of isoprene peroxy radicals (ISOPO$_2$) is substantially affected by the NO and HO$_2$ concentrations, which modulates the strength of the IEPOX-SOA pathway, consistent with observations in different regions (Hu et al., 2015). In the HO$_2$ dominant regions (a), most ISOPO$_2$ reacts with HO$_2$ to produce ISOPOOH and later IEPOX with a yield

of 21.4%. On the other hand, the IEPOX yield is lower (7.5% here) for regions where the NO pathway is dominant (b). An opposite tendency is calculated for an IEPOX-SOA yield from IEPOX, implying the



non-linear chemistry by various factors. The IEPOX-SOA yield from IEPOX are 17.3% (3.7/21.4) and

25.3% (1.9/7.5), respectively for (a) Amazon and (b) Beijing based on GEOS-Chem model calculations,

which can be mainly explained by the higher available aerosol surface area in Beijing compared to

Amazon.

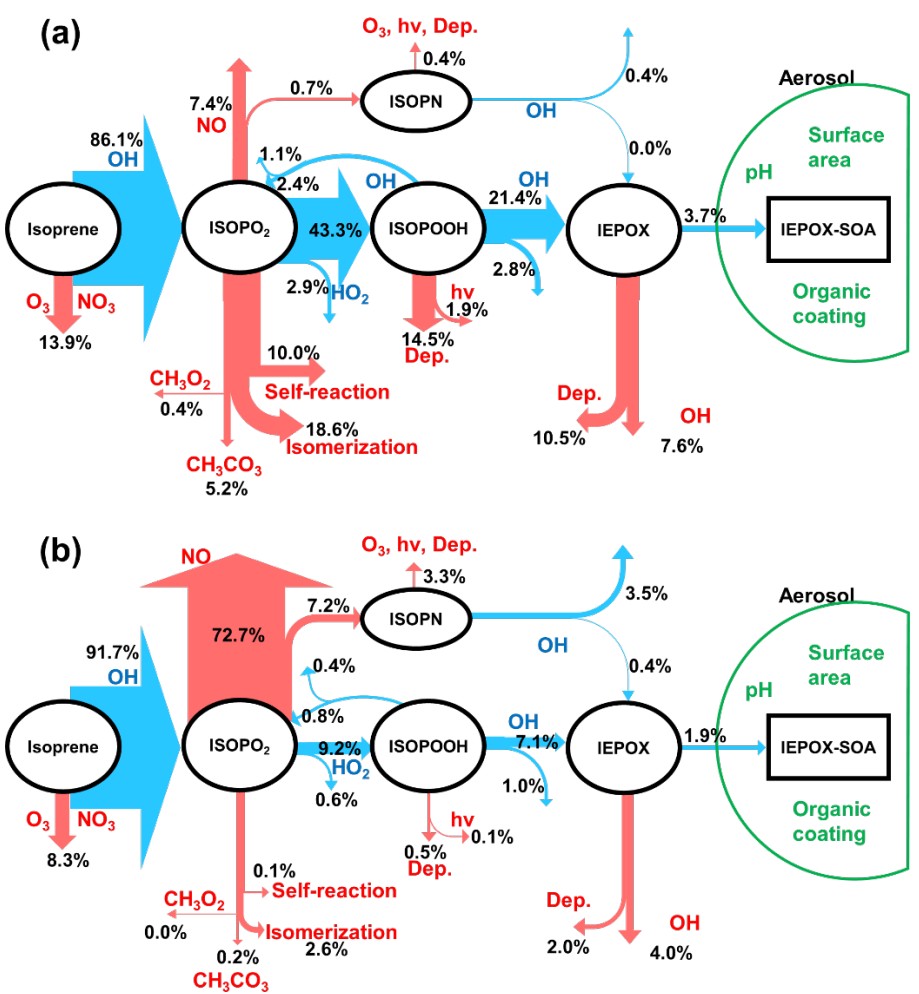

*Figure 1. Schematic diagrams of IEPOX-SOA chemistry for (a) HO₂ and (b) NO dominant regions. Blue arrows indicate IEPOX-SOA formation pathways and red arrows represent other chemical pathways that*
*do not form significant IEPOX-SOA. Values are averaged molar yields relative to isoprene from GEOS-Chem v11-02-rc results over the Amazon (as an example of HO₂ dominant conditions, 10°S – 0°S, 70°W – 60°W) and Beijing (as an example of NO dominant conditions, 35°N – 45°N, 110°E – 120°E) from July 2013 to June 2014. We note that Beijing is located in a region with typically low isoprene emissions, so the appreciable yield of IEPOX-SOA will still result in small ambient concentrations.*



Marais et al. (2016) reported that the model with the explicit irreversible uptake of isoprene SOA precursors to aqueous aerosols coupled to detailed gas-phase chemistry predicted isoprene SOA better than the default isoprene SOA mechanism based on volatility basis set (VBS) in GEOS-Chem v09-02. The VBS mechanism is based on the reversible partitioning of first-generation semivolatile oxidation products onto pre-existing dry OA (Pye et al., 2010). The default VBS mechanism underestimated the

observed isoprene SOA formation by a factor of 3 over the southeast US in summer, whereas the model with the detailed isoprene chemistry showed a close agreement with the measured aircraft and surface isoprene-derived SOA concentrations.

    The use of increasingly detailed chemistry in models enables realistic prediction of chemical composition in the atmosphere, but it is limited by the prohibiting computational cost. As a result, most

of the models participating in the 5$^{th}$ phase of the Coupled Model Intercomparison Project (CMIP5) (Taylor et al., 2011), which provided results for the recent IPCC report (Stocker et al., 2013), used very simplified approaches, such as assuming a constant fraction of emissions to occur as non-volatile SOA (Tsigaridis and Kanakidou, 2018). These simplified approaches were also used in many models participating in the recent AeroCom intercomparison study of OA (Tsigaridis et al., 2014). The modeling

community has tried to improve computational efficiency by condensing complex VBS schemes into simpler ones (Shrivastava et al., 2011; Koo et al., 2014) or by developing empirical parameterizations based on field observations (Hodzic and Jimenez, 2011; Kim et al., 2015). In order to avoid the extra computational cost of the full isoprene mechanism, GEOS-Chem v11-02-rc includes a fixed 3% yield of SOA from isoprene emission for most model applications based on the study by Kim et al. (2015) and

confirmed by the study with the explicit isoprene SOA mechanism in Marais et al. (2016). However, the 3% yield was derived from the measurements over the southeast US during summer in 2013 (Marais et al., 2016), but the explicit isoprene SOA mechanism estimated wide range of SOA yields (3 – 13%) in different years (Marais et al., 2017), implying that isoprene SOA yields could be different under different physico-chemical environments in other regions and time periods (Hu et al., 2015).

In this study, we develop IEPOX-SOA parameterizations based on approximate analytical solutions of the relevant portion of the isoprene chemical mechanism supplemented with numerical fitting. First, a box model is used to develop and evaluate the parameterizations. We then implement the

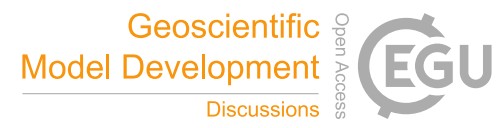

parameterizations into GEOS-Chem and compare the results against those from the explicit irreversible uptake of isoprene SOA precursors to aqueous aerosols coupled to detailed gas-phase chemistry, the default fixed 3% yield, and the VBS scheme. We investigate the performance and limitations of the new parameterizations in terms of global tropospheric concentrations, vertical profiles, and burdens. Our methods substantially reduce computational cost of the explicit isoprene SOA mechanism and provide a much-improved simulation compared to the fixed 3% yield and the VBS parameterizations.

## 2 Global model description

### 2.1. General

We used the GEOS-Chem (v11-02-rc) global 3-D chemical transport model (Bey et al., 2001) to run the parameterizations described in Sect. 3, as well as the explicit isoprene SOA mechanism, fixed 3% yield, and VBS schemes. The model was driven by Goddard Earth Observing System – Forward Processing (GEOS-FP) assimilated meteorological data from the NASA Global Modeling and Assimilation Office (GMAO) for a year (July 2013 to June 2014) with a spin-up time of two months. Winds, temperature, precipitation, and other meteorological variables are provided at $0.3125°$ (longitude) $\times 0.25°$ (latitude) and regridded to $2.5°$ (longitude) $\times 2°$ (latitude) for computational efficiency. GEOS-Chem simulates gas-phase chemistry and aerosol formation including sulfate, ammonium, nitrate (Park et al., 2006), black carbon (Park et al., 2003), OA (Pye et al., 2010; Kim et al., 2015; Marais et al., 2016), sea salt (Jaeglé et al., 2011), and dust (Fairlie et al., 2007). Gas-particle partitioning of inorganic aerosols and aerosol pH are computed with the ISORROPIA II thermodynamic model (Fountoukis and Nenes, 2007; Pye et al., 2009).

### 2.2. Update to the full mechanism of IEPOX-SOA uptake

We updated the standard mechanism and code of GEOS-Chem v11-02-rc to include two recent scientific findings influencing IEPOX-SOA uptake rate. First, we considered organic coating effects when we calculated reactive IEPOX uptake by assuming core (inorganic) – shell (organic) mixing state (Gaston et al., 2014a). Standard GEOS-Chem assumes no organic coating; only the surface area of sulfate. We

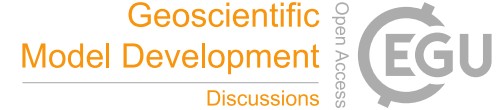



updated the model to include suppression of IEPOX reactive uptake by the organic coating, and to use

available surface area of the total sulfate-ammonium-nitrate-organic aerosols mixture at a given relative humidity with hygroscopic growth factors. As a result, the inclusion of coating reduces the IEPOX reactive uptake coefficient ($\gamma$) (Anttila et al., 2006; Gaston et al., 2014b), but the increased aerosol surface area increases the first order IEPOX loss rate on aerosols as shown in the Eq. (1). The net effect increases the first order IEPOX uptake rate as organic mass increases (Fig. S1). The inclusion of this effect might

result in the overestimation of IEPOX-SOA concentration of GEOS-Chem because we did not alter the Henry's law constant of IEPOX (H = 1.7 x $10^7$ M atm$^{-1}$) or mass accommodation coefficient ($\alpha$ = 0.1) from those used in GEOS-Chem v11-02-rc, which showed the similar IEPOX-SOA concentrations compared to measurements (Marais, 2018). However, it does not affect our conclusions because we compare parameterizations against the explicit isoprene SOA mechanism, not against measurements.

Updating the parameterizations developed here with more accurate values of H or $\alpha$ determined in future literature studies would be trivial. The equation for the uptake rate of IEPOX to form IEPOX-SOA is:

$$\text{IEPOX uptake rate} = \frac{S_a}{\dfrac{r_a}{D_g} + \dfrac{4}{\gamma \times v_{\text{mms}}}} \tag{1}$$

$S_a$ is the wet aerosol surface area on which IEPOX can be taken up (cm$^2$ cm$^{-3}$), $r_a$ is the wet aerosol radius (cm), $D_g$ is gas-phase diffusion coefficient of IEPOX (cm$^2$ s$^{-1}$), and $v_{mms}$ is the mean molecular speed (cm

s$^{-1}$) of gas-phase IEPOX.

Second, we calculate the submicron aerosol pH without sea salt based on the results from previous studies (Noble and Prather, 1996; Middlebrook et al., 2003; Hatch et al., 2011; Allen et al., 2015; Guo et al., 2016; Bondy et al., 2018; Murphy et al., 2018), which showed that sea salt aerosols were dominantly externally mixed with sulfate-nitrate-ammonium rather than internally mixed. Therefore, sea salt is not

expected to impact submicron aerosol pH significantly in the real atmosphere. Effects of sea salt on pH and detailed analysis against the aircraft measurements will be separately discussed in Nault et al. (2019, in preparation).

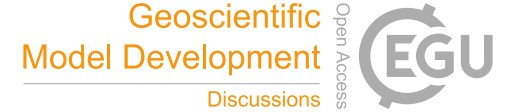

### 2.3. Isoprene SOA simulations

In this section, we briefly describe three different schemes for isoprene SOA simulations used in
GEOS-Chem v11-02-rc: the explicit scheme (Marais et al., 2016), the VBS (Pye et al., 2010), and the
fixed 3% parameterization (Kim et al., 2015). In the explicit scheme, isoprene and its products, and related
processes including chemistry, dry and wet deposition, and transport are explicitly calculated in GEOS-
Chem. IEPOX-SOA formation mechanism is shown in Table S1. Gas-phase concentrations of isoprene,
$ISOPO_2$, ISOPOOH, IEPOX, and isoprene nitrate (ISOPN) are explicitly calculated in every model grid
point. All the species (except for $ISOPO_2$ because of its short life time) are transported in the model. More
detailed information can be found in Marais et al. (2016), with some updates for isomer reactions
described in Sect. 3.1.

The VBS scheme implemented in GEOS-Chem uses six tracers to simulate isoprene SOA, three for
gas-phase and three for aerosol-phase concentrations. This scheme calculates semi-volatile products from
the isoprene + OH reaction and distribute them into three saturation vapor pressure bins ($C* = 1, 10, 100$
$\mu g\ m^{-3}$). These products are partitioned into gas (ISOG1–3 in GEOS-Chem) and aerosol phase (ISOA1–
3 in GEOS-Chem) at every model timestep based on equilibrium partitioning (Pankow, 1994). Dry and
wet deposition are calculated for both gas and aerosol species, with a Henry's law solubility coefficient
of $10^5$ M atm$^{-1}$ (similar to $HNO_3$) for gas species. More detailed description is available in Pye et al.
(2010).

The fixed 3% parameterization applies the fixed 3% mass yield to isoprene emissions to produce two
tracers including the gas-phase SOAP (SOA precursor, with 1.5% mass yield) and the aerosol product
SOAS ("simple" SOA, with the 1.5% yield). The gas-phase tracer SOAP is further aged with a fixed 1-
day conversion timescale to SOAS. There are no losses in the gas-phase for SOAP other than formation
of SOAS.

$$E_{\text{Isoprene}} \times 1.5\% \rightarrow \text{SOAP} \xrightarrow{1\ day} \text{SOAS} \tag{2a}$$

$$E_{\text{Isoprene}} \times 1.5\% \rightarrow \text{SOAS} \tag{2b}$$

Since the fixed 3% and the VBS scheme do not separate IEPOX-SOA from isoprene SOA, we directly
compared isoprene SOA from the VBS and the fixed 3% with the parameterizations developed in Sect. 3.





Because IEPOX-SOA is thought to comprise the dominant fraction of isoprene SOA, we think this assumption will not significantly affect our conclusions. Furthermore, isoprene SOA from the VBS and the fixed 3% parameterizations substantially underestimate the predicted IEPOX-SOA concentrations (Sect. 4), implying that the underestimation will be even larger for total isoprene SOA, if other pathways are significant.

## 3 Parameterization Development

### 3.1. Chemical reactions

We use the explicit isoprene SOA formation mechanism coupled to detailed gas-phase isoprene chemistry from GEOS-Chem v11-02-rc (Yantosca, 2018) as the complete mechanism from which to develop the parameterization. The IEPOX-SOA formation pathway in v11-02-rc is mostly based on Marais et al.
(2016), with updates for inclusion of isomers of ISOPOOH and IEPOX (Bates et al., 2014; St. Clair et al., 2016). As in Marais et al. (2016), we lumped together isomers of the same species to make the resulting parameterizations simpler. Listed in Table S1 are the mechanism used in GEOS-Chem v11-02-rc and the isomer-lumped mechanism, which were used as a starting point for our work. Most reactions forming IEPOX-SOA were included, but we excluded a minor pathway from the isoprene + $NO_3$ reaction, which
contributed only 0.06% of global annual IEPOX production using GEOS-Chem (July 2013 to June 2014). We compared IEPOX-SOA molar yields from isoprene between the isomer-resolved and the isomer-lumped mechanisms for 14,000 different input parameter combinations (using the box model described in Sect. 3.2), which showed nearly identical results (Fig. S2; slope = 1.00 and $R^2$ = 1.00). Hereinafter, we use the word "the full chemistry" or "FULL" to refer to "the explicit IEPOX-SOA formation mechanism
coupled to the detailed gas-phase isoprene chemistry", for brevity.

### 3.2. Box model calculation

We used a box model (KinSim v3.71 in Igor Pro 7.08 (Peng et al., 2015) to simulate IEPOX-SOA concentrations and develop parameterizations. Box model simulations were computed for 10 days with 400 second output timesteps for the complete consumption of isoprene and intermediates. We evaluate
the developed parameterization in Sect. 3.3 by the mechanism over a very wide range of all the key





parameters. We conducted 14,000 box model simulations by varying key species concentrations, aerosol pH and physical properties, temperature, and planetary boundary layer (PBL) height logarithmically over their relevant global tropospheric ranges (Table S2). Aerosol properties are used for the calculation of the IEPOX uptake reaction (R18) (Gaston et al., 2014a, 2014b; Hu et al., 2016). Dry deposition frequencies

(R21-22) were estimated as 2.5 cm s$^{-1}$ / [PBL height] based on measured dry deposition velocity over the southeast United States temperate mixed forest in the summer (Nguyen et al., 2015).

### 3.3. Parameterization 1

We developed three IEPOX-SOA parameterizations based on an approximation of the analytical solution to the chemical mechanism in Table S1. The development of the first parameterization (PAR1) is

described here. First, we divided the IEPOX-SOA formation pathway into four parts:

IEPOX-SOA = $E_{\text{Isoprene}} \times Y_{\text{IEPOX-SOA}}$

$$= E_{\text{Isoprene}} \times f_{\text{Isoprene} \rightarrow \text{ISOPO}_2} \times f_{\text{ISOPO}_2 \rightarrow \text{ISOPOOH}} \times f_{\text{ISOPOOH} \rightarrow \text{IEPOX}} \times f_{\text{IEPOX} \rightarrow \text{IEPOX-SOA}} \quad (3)$$

where *IEPOX-SOA and $E_{Isoprene}$* are the formation rate and emissions of those species [molec. m$^{-2}$ s$^{-1}$]. $Y_{\text{IEPOX-SOA}}$ is the molar yield from isoprene. $f_{A \rightarrow B}$ means the mole fraction of product species *B* formed

upon consumption of precursor species *A*. For example, if $f_{A \rightarrow B}$ is 0.3, 30% of *A* produces *B*, and the remaining 70% of *A* is lost by other chemical reaction pathways. Each fraction can be estimated using the instantaneous reaction rates and species concentrations. For example, the first fraction can be written as:

$$f_{\text{Isoprene} \rightarrow \text{ISOPO}_2} = \frac{k_1 \times [\text{OH}] + k_4 \times [\text{Cl}]}{k_1 \times [\text{OH}] + k_2 \times [O_3] + k_3 \times [\text{NO}_3] + k_4 \times [\text{Cl}]} \quad (4)$$

where $k_n$ represents the reaction rate constant of reaction number *n* in Table S1. Brackets refer to

species concentrations in molec. cm$^{-3}$.

Deriving the second conversion fraction (ISOPO$_2$ → ISOPOOH) in Eq. (3) is not straightforward, due to the ISOPO$_2$ self-reaction (R8). ISOPO$_2$ concentrations change with time and species concentrations. Therefore, we constrained this fraction by performing a numerical fitting method (using the curve fitting analysis tools within Igor Pro) to the output of the box model for the 14,000 independent simulations

discussed above. The result was as follows:





$$f_{\text{ISOPO}_2 \to \text{ISOPOOH}} = Y_5 \times \frac{k_5 \times [\text{HO}_2]}{L_{\text{ISOPO}_2\_others} + L_{\text{ISOPO}_2\_self}} \tag{5a}$$

$$L_{\text{ISOPO}_2\_others} = k_5 \times [\text{HO}_2] + k_6 \times [\text{NO}] + k_7 \times [\text{CH}_3O_2] + k_9 \times [\text{CH}_3CO_3] + k_{10} \tag{5b}$$

$$L_{\text{ISOPO}_2\_self} = C_1 \times \left( 1 - \left( \frac{L_{\text{ISOPO}_2\_others}{}^{C_2}}{L_{\text{ISOPO}_2\_others}{}^{C_2} + C_3{}^{C_2}} \right) \right) \tag{5c}$$

Where $C_1 = 1.207 \times 10^{-2} - 1.048 \times 10^{-2} \times \exp(-2260 \times [P_{ISOPO_2}])$, $C_2 = 1.24$, and $C_3 = 3.667 \times 10^{-2} -$

3.149 $\times 10^{-2} \times \exp(-2411 \times [P_{ISOPO_2}])$. $Y_n$ means the product yield parameter of reaction number $n$ in

Table S1 (i.e., $Y_5 = 0.937$). If the number of products of interest in a single reaction is larger than 1, we

used the notation $Y_{n,m}$ where $n$ denotes the reaction and $m$ the product number (see Eq. (8) below and R6

in Table S1 for example). $P_{ISOPO_2}$ is the production frequency term of ISOPO$_2$ from isoprene ($= k_1 \times$

$[OH] + k_4 \times [Cl]$). The need for this numerical fitting function reflects the fact that ISOPO$_2$

concentration is affected by the loss frequency ($L_{ISOPO_2\_others}$) and the production frequency ($P_{ISOPO_2}$) of

ISOPO$_2$.

The third conversion fraction in Eq. (3) includes the regeneration of ISOPO$_2$ from ISOPOOH (R11).

To consider this regeneration, the resulting IEPOX formation rate $f_{isoprene \to IEPOX,HO_2}$ [molec. m$^{-2}$ s$^{-1}$] from

ISOPO$_2$ + HO$_2$ pathway can be calculated using a geometric series:

$$f_{\text{Isoprene} \to \text{IEPOX},\text{HO}_2} = f_{\text{Isoprene} \to \text{ISOPO}_2} \times f_{\text{ISOPO}_2 \to \text{ISOPOOH}} \times f_{\text{ISOPOOH} \to \text{IEPOX}}$$
$$+ f_{\text{Isoprene} \to \text{ISOPO}_2} \times f_{\text{ISOPO}_2 \to \text{ISOPOOH}} \times f_{\text{ISOPOOH} \to \text{ISOPO}_2} \times f_{\text{ISOPO}_2 \to \text{ISOPOOH}} \times f_{\text{ISOPOOH} \to \text{IEPOX}}$$
$$+ \dots \tag{6a}$$

$$f_{\text{ISOPOOH} \to \text{IEPOX}} = Y_{12} \times \frac{k_{12} \times [\text{OH}]}{k_{11} \times [\text{OH}] + k_{12} \times [\text{OH}] + k_{21} + k_{22}} \tag{6b}$$

$$f_{\text{ISOPOOH} \to \text{ISOPO}_2} = Y_{11} \times \frac{k_{11} \times [\text{OH}]}{k_{11} \times [\text{OH}] + k_{12} \times [\text{OH}] + k_{21} + k_{22}} \tag{6c}$$

Equation (6a) can be solved as $f_{isoprene \to IEPOX,HO_2} = a / (1 - r)$, where

$$a = f_{\text{Isoprene} \to \text{ISOPO}_2} \times f_{\text{ISOPO}_2 \to \text{ISOPOOH}} \times f_{\text{ISOPOOH} \to \text{IEPOX}} \tag{6d}$$

$$r = f_{\text{ISOPOOH} \to \text{ISOPO}_2} \times f_{\text{ISOPO}_2 \to \text{ISOPOOH}} \tag{6e}$$

Finally, the fourth function can be calculated as:



$$f_{\text{IEPOX}\rightarrow\text{IEPOX-SOA}} = \frac{k_{18}}{k_{17} \times [\text{OH}] + k_{18} + k_{23}} \tag{7}$$

Analogously, the IEPOX formation rate from the ISOPO$_2$ + NO pathway can be calculated as follows:

$$f_{\text{Isoprene}\rightarrow\text{IEPOX,NO}} = \frac{k_1 \times [\text{OH}] + k_4 \times [\text{Cl}]}{k_1 \times [\text{OH}] + k_2 \times [O_3] + k_3 \times [\text{NO}_3] + k_4 \times [\text{Cl}]} \times \{\frac{k_6 \times [\text{NO}]}{L_{\text{ISOPO}_2\_others} + L_{\text{ISOPO}_2\_self}} \times$$

$$(Y_{6,1} \times Y_{13} \times \frac{k_{13} \times [\text{OH}]}{k_{13} \times [\text{OH}] + k_{15} \times [O_3]} + Y_{6,2} \times Y_{14} \times \frac{k_{14} \times [\text{OH}]}{k_{14} \times [\text{OH}] + k_{16} \times [O_3]})\} \tag{8}$$

With both HO$_2$ and NO pathways combined, the IEPOX-SOA yield ($Y_{IEPOX\text{-}SOA}$) is

$$Y_{\text{IEPOX-SOA}} = (f_{\text{isoprene}\rightarrow\text{IEPOX,HO}_2} + f_{\text{Isoprene}\rightarrow\text{IEPOX,NO}}) \times f_{\text{IEPOX}\rightarrow\text{IEPOX-SOA}} \tag{9}$$

From Eq. (9), we can calculate the IEPOX-SOA molar yield with instantaneous meteorological and chemical fields in each grid box. We evaluated this instantaneous IEPOX-SOA molar yield against the calculated IEPOX-SOA yield using the full mechanism with the box model (Fig. S3a). Each point indicates the IEPOX-SOA yield with randomly selected input variables in the parameter space shown in Table S2. We confirmed that the yield from Eq. (9) very accurately regenerated the simulated yield from

the full mechanism with the box model (Fig. S3).

Equation (9) gives the instantaneous yield if all the reactions were extremely fast, but it takes time to produce IEPOX-SOA in the full chemistry model as well as in the real atmosphere. As a result, if the yield from Eq. (9) is used for making IEPOX-SOA, chemical transport models would likely overestimate IEPOX-SOA concentrations locally in isoprene-emitting areas because of its too rapid formation. To

simulate the formation of IEPOX-SOA with a realistic timescale, we introduced a single gas-phase intermediate, similarly to the 3% parameterization in GEOS-Chem v11-02-rc. The gas-phase intermediate is then converted to IEPOX-SOA with a first order timescale that depends on the local conditions. The final form of parameterization PAR1 is:

$$E_{\text{Isoprene}} \times Y_{\text{IEPOX-SOA}} \rightarrow \text{SOAP} \xrightarrow{\tau} \text{IEPOX-SOA} \tag{10}$$

*SOAP* stands for the gas-phase precursor of IEPOX-SOA (using the same terminology as in the 3% parameterization in GEOS-Chem), and $\tau$ is the formation timescale. *SOAP* represents the lumped species of isoprene, ISOPOOH, and IEPOX, and it undergoes wet deposition with the effective Henry's law solubility coefficient of $10^5$ M atm$^{-1}$ (the value used for the gas-phase semivolatile products of isoprene

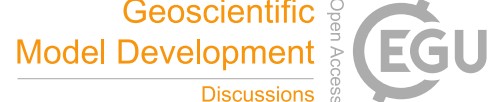



SOA simulated by the VBS in GEOS-Chem). Dry deposition of *SOAP* was not simulated in GEOS-Chem,

because dry deposition of intermediate species was already included in the parameterization (R22 and

R23). On the other hand, *SOAP* in the 3% parameterization is not dry or wet deposited, as described in

Sect. 2.3. (Kim et al., 2015; Yantosca, 2016). IEPOX-SOA formation is calculated at each timestep ($\Delta t$)

in the model as follows:

$$\text{IEPOX-SOA(t+}\Delta\text{t)} = \text{IEPOX-SOA}(t) + \left\{1 - \exp\left(-\frac{\Delta t}{\tau}\right)\right\} \times \text{SOAP}(t) \tag{11}$$

We conducted numerical fitting to calculate the value of $\tau$, due to the fact that many processes in the

mechanism can affect the formation timescale of IEPOX-SOA. The best fitting results were obtained from

Hill equation formulas with the loss rates of different precursors as shown in Eq. (12) below.

$$\tau = C_0 + C_1 \times \frac{L_{\text{ISOP}}{}^{C_2}}{L_{\text{ISOP}}{}^{C_2} + C_3{}^{C_2}} + C_4 \times \frac{L_{\text{ISOPOOH}}{}^{C_5}}{L_{\text{ISOPOOH}}{}^{C_5} + C_6{}^{C_5}} \times F + C_7 \times \frac{L_{\text{ISOPN}}{}^{C_8}}{L_{\text{ISOPN}}{}^{C_8} + C_9{}^{C_8}} \times (1 - F)$$

$$+ C_{10} * \frac{L_{\text{IEPOX}}{}^{C_{11}}}{L_{\text{IEPOX}}{}^{C_{11}} + C_{12}{}^{C_{11}}} \tag{12a}$$

$$F = C_{13} + C_{14} \times \exp\left(-C_{15} \times \frac{P_{\text{ISOPOOH}}}{L_{\text{ISOPO}_2}}\right) + C_{16} \times \exp\left(-C_{17} \times \frac{P_{\text{ISOPN}}}{L_{\text{ISOPO}_2}}\right) \tag{12b}$$

Where $L$ stands for the loss frequency of a species [s$^{-1}$], and $P$ represents the production frequency of a

species [s$^{-1}$]. Constants are listed in Table S3. Equation (12a) has five parts – constant ($C_0$), isoprene

(ISOP) loss ($C_1$–$C_3$), ISOPOOH loss ($C_4$–$C_6$), ISOPN loss ($C_7$–$C_9$), and IEPOX loss ($C_{10}$–$C_{12}$). All

precursor loss rates affect the formation timescale except for ISOPO$_2$ loss. The loss rate of ISOPO$_2$ is very

fast, therefore, it rarely influences the formation timescale of IEPOX-SOA. There are two different

ISOPO$_2$ loss pathways leading to IEPOX. We included these pathways by calculating Eq. (12b) with

HO$_2$/NO branching ratio from ISOPO$_2$ oxidation. As shown in Fig. S3b, the formation timescale by box

model was generally well captured by the parameterization over the entire input parameter space (slope

= 0.98 and R$^2$ = 0.98).

**3.4. Parameterizations 2 and 3**


PAR1 showed some limitations in performance (discussed below), which were related to the calculation

of $Y_{IEPOX\text{-}SOA}$ based on the local conditions when isoprene is emitted. Since the time to form and uptake



IEPOX can be significant, and some parametric dependences are quite nonlinear (especially for IEPOX uptake), this approximation can result in some deviations between the parameterization and the full

chemistry since the local conditions at the time of IEPOX uptake may be different than those at the time of isoprene emission. To address this problem and improve performance, a modified second parameterization (PAR2) was developed, where the gas-phase IEPOX($g$) yield is calculated with the local conditions at the point of isoprene emissions, while the IEPOX uptake to form IEPOX-SOA is calculated explicitly using Eq. (14). $Y_{IEPOX}$ was calculated from Eq. (9), by eliminating $f_{IEPOX \rightarrow IEPOX-SOA}$ from the right

side of the equation. The form of PAR2 is:

$$E_{\text{Isoprene}} \times Y_{\text{IEPOX}} \rightarrow \text{IEPOX} \xrightarrow{f_{\text{IEPOX} \rightarrow \text{IEPOX-SOA}}} \text{IEPOX-SOA} \qquad (13)$$

IEPOX-SOA formation is calculated at each timestep ($\Delta t$) in the model as follows:

$$\text{IEPOX-SOA(t+}\Delta\text{t)} = \text{IEPOX-SOA}(t) + \{1 - \exp(-\Delta t \times (k_{17} \times [\text{OH}] + k_{18} + k_{23})\}$$

$$\times \text{IEPOX}(t) \times \frac{k_{18}}{k_{17} \times [\text{OH}] + k_{18} + k_{23}} \qquad (14)$$

PAR2 effectively replaces the generic SOAP gas-phase intermediate of PAR1 with a chemically-meaningful gas-phase intermediate (IEPOX).

    Because IEPOX is formed immediately after isoprene emission in PAR2, it can result in an overestimated IEPOX concentrations since the gas-phase chemistry has a limited rate. Therefore, we developed a 3$^{rd}$ parameterization (PAR3) by modifying PAR2 by representing the formation timescale for IEPOX by adding a second intermediate:


$$E_{\text{Isoprene}} \times Y_{\text{IEPOX}} \rightarrow \text{SOAP}_I \xrightarrow{\tau_I} \text{IEPOX} \xrightarrow{f_{\text{IEPOX} \rightarrow \text{IEPOX-SOA}}} \text{IEPOX-SOA} \qquad (15)$$

Where $\tau_I$ is the formation timescale of IEPOX, which is calculated using the equation below.

$$\tau_I = C_0 + C_1 \times \frac{L_{\text{ISOP}}^{C_2}}{L_{\text{ISOP}}^{C_2} + C_3^{C_2}} + C_4 \times \frac{L_{\text{ISOPOOH}}^{C_5}}{L_{\text{ISOPOOH}}^{C_5} + C_6^{C_5}} \times F + C_7 \times \frac{L_{\text{ISOPN}}^{C_8}}{L_{\text{ISOPN}}^{C_8} + C_9^{C_8}} \times (1 - F) \quad (16)$$

    The functional form of Eq. (16) is the same as Eq. (12a) but excludes the last term (IEPOX loss). $F$ is

calculated using Eq. (12b) but with different constant values, which are provided in Table S3. Similar to the evaluation of PAR1, $Y_{IEPOX}$ and $\tau_I$ were generally well predicted compared to 14,000 box model simulations (Fig. 2).

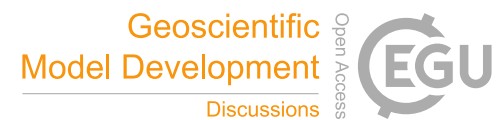

Three parameterizations from Eqs. (10), (13), and (15) were implemented in GEOS-Chem and evaluated in the rest of the paper. For brevity, hereinafter the parameterization using Eq. (10), Eq. (13), and Eq. (15) are referred to simply as "PAR1", "PAR2", and "PAR3", respectively.

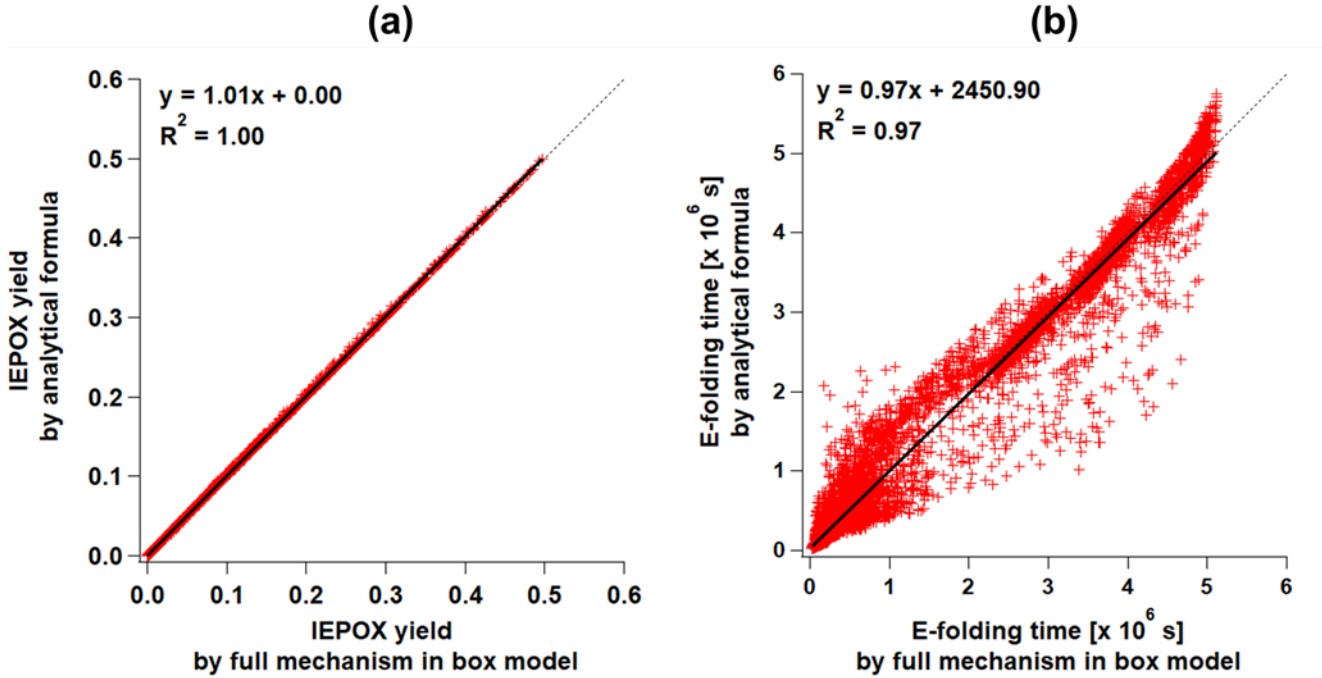

*Figure 2. Scatterplots of the results of parameterization PAR3 (y-axis) versus the full mechanism (x-axis) box model results for (a) IEPOX-SOA molar yield and (b) formation timescale.*

# 4 Results

## 4.1. Full chemistry vs. Parameterizations

Figure 3 shows global annual surface maps of simulated IEPOX-SOA concentrations by using full chemistry and the five parameterizations, while Figure 4 compares the concentrations and burdens. The fixed 3% yield parameterization (FIXED) substantially underestimated IEPOX-SOA concentrations with a slope of 0.34. This underestimation became even worse for IEPOX-SOA concentrations below 0.1 μg m$^{-3}$ where most of the points are located (Fig. S4).





Similar to the 3% parameterization, isoprene SOA concentrations with the VBS were substantially lower than those with the full chemistry and parameterizations. Isoprene SOA ratios of the VBS to the full chemistry were less than 20% except for the source regions (Fig. 3c), because more semi-volatile products can exist in aerosol phase due to high pre-existing aerosol concentrations in source regions. Furthermore, the VBS/Full chemistry ratios were even higher than 1 for anthropogenic source dominant regions (California, western Europe, and Asia), where NO concentrations are high. However, the VBS predicted very low isoprene SOA concentrations in remote regions, leading to a low global burden (Fig. 4c). This dramatic difference came from the fact that the IEPOX-SOA is non-volatile in the full chemistry, but the isoprene SOA is treated as semi-volatile using the partitioning theory in the VBS. The VBS simulated most of the semi-volatile products as gas-phase (tropospheric burden of 237 Gg) rather than aerosol-phase (tropospheric burden of 49 Gg), especially for remote regions where pre-existing aerosol concentrations were low.

PAR1 generally underestimated IEPOX-SOA concentrations compared to the full chemistry simulation (slope = 0.68; $R^2$ =0.9), although with less bias and better skill than the default VBS (slope = 0.29; $R^2$ = 0.45) and the fixed 3% yield simulation (slope = 0.34; $R^2$=0.65). An important driver of the low bias vs. the full chemistry was the diurnal variation of the chemical fields. $Y_{IEPOX\text{-}SOA}$ is calculated in PAR1 using the instantaneous chemical fields at the time of isoprene emission, while in the full chemistry simulation (and in the real atmosphere), some processes proceed at different rates due to the different diurnal variations of key parameters.



**Figure 3.** *Annual mean (July 2013 – June 2014) surface concentrations for IEPOX-SOA as predicted by full chemistry (a). Ratio of parameterized IEPOX-SOA concentrations to the full chemistry case are shown in (b,c,d,e,f).*

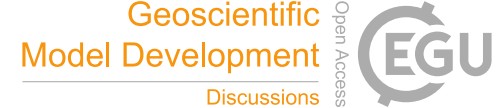

To directly investigate the effect from the diurnal variation of the chemical fields, we used the box model to exclude other factors such as transport and deposition processes. First, we extracted isoprene emissions and chemical/meteorological fields affecting the IEPOX-SOA formation pathway from GEOS-Chem with 30 minutes temporal resolution (equivalent to the chemistry timestep of GEOS-Chem used in this study). Then we averaged global chemical/meteorological fields within the PBL based on local time at each grid point for four major isoprene source regions (the Southeastern United States, Amazon, Central Africa, and Borneo). In this way, we constructed the source regions–averaged diurnal profile of chemical species, temperature, boundary layer height, isoprene emission, and reaction rate constants as inputs of the box model. The underestimation of IEPOX-SOA concentrations by PAR1 also occurred when we calculated IEPOX-SOA with the box model (Fig. 4d). This was caused by the diurnal variation of chemical/meteorological fields, as PAR1 successfully captured the timeseries of IEPOX-SOA when we used constant input values (Fig. S5).

The box model simulation with the source regions–averaged diurnal cycle resulted in similar IEPOX-SOA concentrations between the two parameterizations directly calculating IEPOX (PAR2 and PAR3) and full chemistry (Fig 4d). PAR2 and PAR3 also showed similar global spatial patterns vs. the full chemistry, although they slightly overestimated IEPOX-SOA over source regions (Amazon, Central Africa, and Southeast Asia) (Fig. 3d and 3f).

The different performance between PAR1 and PAR2 or PAR3 was mainly caused by the differing influence of the diurnal variation profiles of chemical fields (Fig. S6). In the full chemistry model, isoprene emission and OH peaked around local noon, but the IEPOX uptake rate peaked around 6 p.m. Because it takes several hours to convert the emitted isoprene to IEPOX in the full chemistry case (Fig. 5a), the IEPOX-SOA yield calculated at the time of isoprene emission (in PAR1) could underestimate the real IEPOX-SOA yield. For example, the instantaneous IEPOX-SOA yield using both isoprene emission and IEPOX condensation rates at noon is lower than the yield calculated using the isoprene emission rate at 12 p.m. and the IEPOX condensation rate at 6 p.m, when each process peaks, explaining most of the underestimation in PAR1. Contrary to PAR1, which calculated IEPOX-SOA yield at the time of isoprene emission, PAR2 and PAR3 did not show a global underestimation because they only calculated IEPOX yield at the time of isoprene emission, and then simulated the IEPOX condensation rate explicitly. The



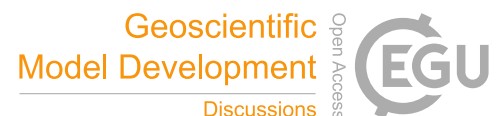
3% parameterization underestimated IEPOX-SOA globally except for Amazon, India, and Western Europe.

*Figure 4. (a) Scatterplots of parameterized (y-axis) versus full chemistry IEPOX-SOA (x-axis) concentrations within the troposphere for July 2013 – June 2014. Each point represents monthly averaged*





*model grid value of IEPOX-SOA concentration. Colors represent the density of points, where densities were calculated by dividing x and y axis ranges into 100 by 100 grid cells. (b) Vertical profiles of global annual mean average IEPOX-SOA concentrations. The vertical locations of the markers indicate the mid levels of the vertical grid boxes in GEOS-Chem. (c) Timeseries of global tropospheric burdens of IEPOX-SOA [Gg]. (d) Timeseries of IEPOX-SOA concentrations simulated by the box model. The VBS was not calculated with the box model, as it requires additional partitioning calculation with pre-existing aerosols, which are calculated online in GEOS-Chem. Input chemical/meteorological fields were averaged from GEOS-Chem results for  four major isoprene source regions [the Southeastern United States: 30°N – 40°N, 100°W – 80°W, Amazon: 10°S – 0°S, 70°W – 60°W, Central Africa: 5°N – 15°N, 10°E – 30°E, Borneo: 5°S – 5°N, 105°E – 120°E]. Input values represent annual mean values, which were calculated by using the first two days of each month model outputs at 30 minutes interval averaged within the PBL.*




*Figure 5. Global budget analysis of IEPOX-SOA formation from isoprene on a total annual mean basis (July 2013 – June 2014). Black arrows with numbers show the IEPOX-SOA formation pathways. Two numbers are shown if the loss amount of reactant differs from the production amount of product (underline italic), which are caused by the different molecular weights and product yields. Chemical losses that are not leading to IEPOX-SOA formation are shown in red arrows. Dry and wet deposition amounts are presented in green and purple arrows, respectively. Tropospheric burdens are given in brackets if species is explicitly simulated in the model. Blue circles are used for species that are explicitly simulated in each case.*




Parameterizations using chemical fields (PAR1, PAR2, and PAR3) captured the variability of IEPOX-
SOA well with $R^2$ values of 0.88–0.93. PAR3 always showed the best $R^2$ and slopes in terms of not only annual mean (Fig. 4a) but also monthly mean evaluation (Fig. S7), due to the fact that the structure of PAR3 was closer to that of full chemistry compared to other parameterizations. PAR3 requires three tracers and has a slightly higher computational cost than PAR1 and PAR2 that need two tracers to simulate IEPOX-SOA (Table 1).

In terms of vertical profiles (Fig. 4b), PAR2 and PAR3 again showed the best results, although these parameterizations slightly overestimated surface concentrations and underestimated upper troposphere concentrations. The slightly shorter average lifetime of IEPOX-SOA by PAR2/PAR3 (4.6–4.7 days) vs. full chemistry (5.2 days) (Fig. 5) resulted in slightly reduced upward transport. On the other hand, PAR1, the VBS, and the 3% yield substantially underestimated both surface and free tropospheric concentrations.

The annual mean global tropospheric burden of IEPOX-SOA by full chemistry was 223 Gg, vs. 123, 222, 201, 49, and 82 Gg for PAR1, PAR2, PAR3, the VBS, and the 3%, respectively. Global IEPOX-SOA burdens of PAR2 and PAR3 were within ~10% of IEPOX-SOA burden simulated by full chemistry. Furthermore, we found that PAR2 and PAR3 showed similar monthly variations for global burdens; as well as, absolute values compared to full chemistry (Fig. 4c).

We calculated the annual mean global budgets of IEPOX-SOA simulated by the full chemistry and the parameterizations developed in this study (Fig. 5). Generally, each term is of the same order, with some differences in some cases, which are mainly due to the diurnal variation of the chemical fields. For example, the isoprene loss by $O_3$ and $NO_3$ was 21 Tg yr$^{-1}$ for the full chemistry, but this loss was reduced to 10 Tg yr$^{-1}$ in our parameterizations. Because $NO_3$ concentration was very low during the daytime when
isoprene was emitted (Fig. S6), our parameterizations using the instantaneous yield applied to isoprene emission underestimated isoprene loss by $NO_3$. On the other hand, $ISOPO_2$ loss was higher in our parameterizations (134 Tg yr$^{-1}$) than in the full chemistry (84 Tg yr$^{-1}$) because chemical species affecting $ISOPO_2$ loss ($CH_3O_2$ and $CH_3CO_3$ in Fig. S6) had similar diurnal variation patterns compared to the isoprene emission.

Although there were some differences between the results of the parameterizations and the full chemistry above, the parameterizations generally showed similar source and sink values compared to the

full chemistry. The full chemistry showed annual production of 153 Tg yr$^{-1}$ ISOPOOH, which was similar

to the value estimated by the parameterizations (141 Tg yr$^{-1}$). That was also the case for the annual

production of IEPOX (78 Tg yr$^{-1}$ vs. 70 Tg yr$^{-1}$). Results in Fig. 5 imply that chemical reaction-based

parameterizations can capture global budgets of IEPOX-SOA chemistry with reasonable accuracy

without explicit calculation of all intermediates. Furthermore, we found that the flux from IEPOX (or

SOAP) to IEPOX-SOA was important for IEPOX-SOA simulation capability. For example, the flux from

IEPOX to IEPOX-SOA in PAR3 was 16 Tg yr$^{-1}$, which was same to the flux (16 Tg yr$^{-1}$) in the full

chemistry, and PAR3 showed the best results. On the other hand, the production of IEPOX-SOA was 10

Tg yr$^{-1}$ in PAR1, which was the main reason for the IEPOX-SOA underestimation in that case.

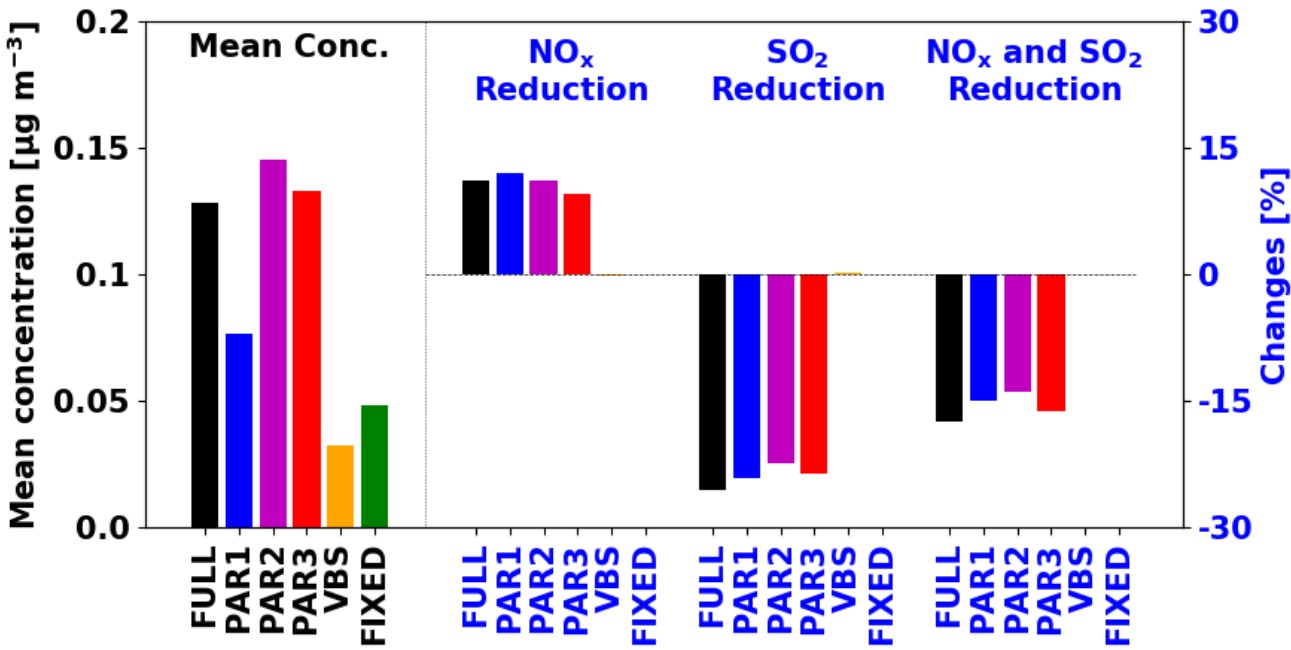

*Figure 6. Global PBL averaged IEPOX-SOA concentrations (left, black) and the concentration changes with anthropogenic emission reductions (right, blue) for July – August 2013. The anthropogenic*
*emissions were decreased by 50% for each sensitivity case.*





Finally, we investigated the effects of anthropogenic emission reductions on the simulated IEPOX-SOA concentrations. We conducted additional sensitivity tests for two months by reducing $NO_x$ and $SO_2$ emissions by 50%. New parameterizations (PAR1–3) showed similar sensitivities to the full chemistry case, but the VBS and fixed 3% parameterizations did not reproduce changes relative to emission reductions (Fig. 6). Isoprene SOA concentrations by the fixed 3% parameterizations remain the same because they are using the constant yield.

The VBS showed negligible sensitivities (less than 0.3%). For the VBS, changes in the rate of oxidation of isoprene is the most important factor that can affect isoprene SOA changes. We found that OH concentrations were decreased in the $NO_x$ reduction case (Fig. S8a). However, isoprene concentrations were increased (Fig. S8b) due to the reduced oxidant fields affecting isoprene loss (OH, $O_3$, and $NO_3$), because the chemical loss is the only pathway for isoprene loss (i.e. no isoprene is lost by dry and wet deposition) and isoprene emissions are unaffected. As a result, the initial rate oxidation of isoprene (rate constant x [isoprene] x [OH]) did not show the significant changes (Fig. S8d), as is also observed for isoprene SOA (Fig. S8f).

However, in the explicit full chemistry, for the sensitivity case of $NO_x$ emission reduction, the contribution of $HO_2$ pathway was increased compared to the NO pathway, making more IEPOX and IEPOX-SOA. The reduced sulfate aerosol caused by the $SO_2$ emission reduction increases aerosol pH and decreases available aerosol surface area, which eventually decreases IEPOX reactive uptake. New parameterizations successfully captured these tendencies, indicating that they will be much more accurate in simulating the response of isoprene SOA to different scenarios, such as the response to future climates or anthropogenic emission reduction scenarios.

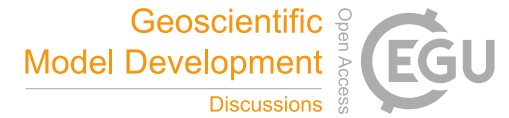



*Table 1. Computational time estimation for the simulation of IEPOX-SOA using the full chemistry and parameterization cases in the box model and GEOS-Chem. The box model results are mean values of 1,000 simulations based on 5-days integration time. The VBS was not simulated in the box model, because the VBS requires the partitioning calculation with pre-existing aerosol concentrations, which are not available in the box model, and are calculated online in GEOS-Chem. For GEOS-Chem, values were*
*based on 7-days simulation using 32 cores on NCAR Cheyenne machine. The Gprof performance analysis tool was used to calculate how much time was spent in subroutines with Intel Fortran Compiler 17.0.1 with '-p' option. Values were estimated by multiplying the total time spent in each process by the contribution of related reactions/species for each case, except for time estimates for chemistry of parameterizations[1]. For example, transport time in full chemistry was calculated by multiplying 2978 s*
*(total transport time in Table S4) by 10 (Total number of the full chemistry species) / 173 (Total number of advected species).*

| | Box model [s] | GEOS-Chem [s] | | | | |
|---|---|---|---|---|---|---|
| | Chemistry | Chemistry | Transport | Dry deposition | Wet deposition | Total |
| FULL | 1.5285 | 559 | 172 | 30 | 380 | 1141 |
| VBS | - | 7 | 120 | 20 | 253 | 400 |
| PAR1 | 0.0028[1] | 47 | 34 | 7 | 84 | 172 |
| PAR2 | 0.0023[1] | 13 | 34 | 7 | 84 | 138 |
| PAR3 | 0.0028[1] | 48 | 52 | 7 | 127 | 234 |
| FIX | 0.0012[1] | 1 | 34 | 3 | 42 | 80 |

## 4.2. Computational time estimation

We estimated computational time related with IEPOX-SOA simulation for the full chemistry and the
different parameterizations. The box model was used for estimating the time needed for chemistry calculation using chemical reactions and dry depositions in Table S1. All the parameterizations showed much faster integration time compared to the full chemistry.

For estimation within GEOS-Chem, we used the Gprof function profiling program and categorized the results according to four major processes (chemistry, transport, dry deposition, and wet deposition), as
shown in Table 1. One of main advantage using a function profiling program is that all of timings are estimated at once without need for multiple simulations. Because model computational time varies



between individual executions even for the same machine and code (Philip et al., 2016), and because we examined a minority (IEPOX-SOA chemistry) of total GEOS-Chem model reactions, computational time estimation using multiple runs can lead to significant errors.

Our parameterizations (PAR1-3) reduced the computational time by factors ~5 and ~2 compared to the full chemistry and the VBS, respectively. There was a factor of two difference among parameterizations due to two main reasons. First, the difference between PAR1 and PAR2 arose from the additional calculation of formation timescale in PAR1 (Eq. 12). Second, the number of species was a key factor making difference between PAR1 (2 species) and PAR3 (3 species). The 3% showed the best efficiency—

the cost of the 3% case was ~2–3 times less than those of the PAR1–3, given its simplest structure.

When using GEOS-Chem, the full chemistry can still be chosen if computational cost is not important or the detailed gas-phase chemical reactions are needed. Our developed parameterizations (PAR1–3) can be useful for researchers who are not interested in the details of isoprene SOA, but who still want to have realistic aerosol concentrations in their simulations. PAR3 adds significant accuracy compared to the 3%

yield GEOS-Chem default for limited additional cost. The VBS requires more computational cost than all of parameterizations while being less accurate, and we recommend against its use in future simulations. Although we have used GEOS-Chem as a convenient development platform, the parameterizations may be especially useful for climate models for long-term simulations using other codes.

**5 Conclusions**

IEPOX-SOA is thought to dominate the contribution of isoprene to SOA, but it is formed by complex multiphase chemistry which cannot be accurately simulated by the commonly used lumped volatility-basis-set or fixed yield SOA schemes. A detailed isoprene chemistry mechanism has been recently developed and implemented in some models, and recent studies have found good agreement between

observed and simulated IEPOX-SOA concentrations. However, the detailed chemistry requires higher computational cost than the lumped SOA schemes, which may not be applicable for long-term multi-scenario simulations in climate and similar models. The likely addition of other explicit SOA mechanisms as knowledge improves in the future would exacerbate this problem.



Here we developed parameterization methods to enable accurate yet fast IEPOX-SOA formation for
climate model applications that mostly require having the correct SOA mass, spatio-temporal distribution, and response to changes in important precursors, for accurate calculations of the aerosol radiative effects. First, we developed a method to calculate the yield of IEPOX-SOA from isoprene emissions based on an approximate analytical solution of the full mechanism. Numerical fitting to box model results was introduced when the reaction could not be directly implemented for yield calculation. Formation
timescales of key products were also used to more accurately represent the characteristic time of formation of IEPOX-SOA. Therefore, our parameterizations used two (PAR1 and PAR2) or three tracers (PAR3) to simulate IEPOX-SOA without the full chemical mechanism.

The parameterizations (especially PAR2 and PAR3) generally captured the spatial and temporal variations of IEPOX-SOA including sources, sinks, burdens, surface concentrations, and vertical profiles.
Furthermore, the parameterizations showed better performance and lower computational cost compared to the current fixed yield or VBS schemes in GEOS-Chem. Therefore, these parameterizations can be used for more accurate predictions of surface concentrations; as well as, climate effects such as direct radiative forcing calculation.

The parameterizations can be easily updated if new values of key parameters are adopted by the
community (e.g. the Henry's law constant of IEPOX). The differences between the parameterizations and the full chemistry were mostly explained by non-linear effects due to the diurnal variation of chemical/meteorological fields, which cannot be captured without additional complexity. One caveat is that some climate models use monthly mean fields of VOCs and oxidants. Because the diurnal variation was found to be important for accurate predictions of IEPOX-SOA, this may reduce the accuracy of the
results for such models. We recommend that climate models account for diurnal variations for each chemical field in order to obtain more accurate IEPOX-SOA concentrations.

Detailed mechanistic studies in the laboratory, often aided by new mass spectrometry instrumentation with higher molecular detail, are leading to the development of many detailed SOA mechanisms, which will challenge global and especially climate models with their increased computational cost. The method
developed in this study can be used to simplify other SOA mechanisms, allowing more accurate SOA simulations while limiting computational cost.




*Code and Data Availability.* The KinSim box model will be publicly available on the web (http://cires1.colorado.edu/jimenez-group/wiki/index.php/Analysis_Software#KinSim_Software) but it is currently available upon request (jose.jimenez@colorado.edu). GEOS-Chem v11-02-rc and meteorological data can be downloaded from GEOS-Chem website (http://wiki.seas.harvard.edu/geos-chem/index.php/Downloading_GEOS-Chem_source_code_and_data). GEOS-Chem code modifications for new parameterizations and global model data are available upon request submitted to the corresponding author (jose.jimenez@colorado.edu).

*Author contributions.* JLJ, AH, and LKE designed the research. ZP developed the KinSim box model. WH conducted the IEPOX reactive uptake calculation within Igor Pro. DSJ and EAM conducted global model simulations. BAN conceived the ideas of aerosol pH calculation. PCJ analyzed the IEPOX-SOA data. DSJ, JLJ, and AH developed the parameterizations. DSJ and JLJ wrote the original paper, and all authors contributed to the review and editing of the paper.

*Competing interests.* The authors declare that they have no conflict of interest.

*Acknowledgements.* This publication was supported by US EPA STAR 83587701-0, NOAA NA18OAR4310113, DOE (BER/ASR) DE-SC0016559, NSF AGS-1822664, and the European Research Council (grant No. 819169). It has not been formally reviewed by EPA. The views expressed in this document are solely those of the authors and do not necessarily reflect those of the Agency. EPA does not endorse any products or commercial services mentioned in this publication. We thank Prasad Kasibhatla for useful discussions.





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
