# Peer review of "A simplified parameterization of isoprene-epoxydiol-derived secondary organic aerosol (IEPOX-SOA) for global chemistry and climate models: a case study with GEOS-Chem v11-02-rc"

_Geoscientific Model Development, 2019_

## Referee Comment (RC1) · Anonymous Referee #1 · 6 Mar 2019

This manuscript proposed three parameterization methods to reduce the computational cost of simulating Isoprene-derived SOA in GEOS-Chem using the full chemistry model. The parameterization methods were also compared with the volatility-basis-set and fixed yield methods. The results show that the parameterization methods, especially the third one (PAR3), could generally predict the isoprene-derived SOA spatially while reducing the computational cost of using a full chemistry model. The manuscript also suggests that the VBS approach is not accurate in predicting isoprene-derived SOA because the reactive uptake process is the main process for isoprene-derived

[Figure]

SOA formation.

The manuscript has clear logic and it provides a useful and efficient parameterization method to calculate isoprene-derived SOA in global models such as GEOS-Chem. The quality of the manuscript is good and the main argument is valid, thus the manuscript is worth being published in Geoscientific Model Development. My main comments are that it occurred in a few places of the manuscript that the assumptions of the model or the detailed processes are not fully clear. I suggest the authors make appropriate changes to the manuscript in the following sections.

1. Line 80: the author states that Pye et al. 2010 used VBS method to predict isoprene-derived SOA from first-generation products through partitioning. However, in Pye et al. 2013, the reactive uptake aqueous pathway was already incorporated to the VBS method in predicting the isoprene-derived SOA. The new VBS results have been improved and are consistent with field measurements. The author should at least incorporate the latest model improvement approach by Pye et al. in the manuscript when discussing the VBS method. With the latest method incorporating 2-methyltetrol and 2-methylglyceric acid, it is inaccurate to state that "the default VBS mechanism underestimated the observed isoprene SOA formation by a factor of 3 over the southeast US in summer".

2. Line 130-135. The author states that the net effect of the coating effect is to increase the reaction rate of IEPOX with organic coated aerosols. However, both Gaston et al. 2015 and Zhang et al. 2018 show that the uptake coefficient is highly dependent on the diffusion coefficient of IEPOX in the organic layer, the Herny's law of the IEPOX into the organic layer, etc. With different parameters, the resistor model can give drastically different results. The author should explicitly specify what the equations and values were used to calculate the uptake coefficient in Figure S1. Also, why would the uptake coefficient increase at higher organic loading in Figure S1? As shown in Anttila et al., 2006, Gaston et al. 2015, and Zhang et al. 2018, the resistor model will generate monotonically decreasing reactive uptake values as the coating thickness increases.

3. Line 140. I understand that the author is intended to cross compare parameterizations against the full chemistry model for isoprene formation, however it is inaccurate and a bit misleading to state that "Updating the parameterizations developed here with more accurate values of H or $\alpha$ determined in future literature studies would be trivial." The H or $\alpha$ values affect the absolute values of the isoprene-derived SOA significantly, and currently the estimation of the Herny's law constant can vary by two orders of magnitudes. Knowing an accurate value of the Herny's law constant will help bridge the gap between the model and the field measurement, thus these values are not trivial. I suggest the author revise this sentence.

4. Please remove the Nault et al. in prep. citation as unpublished work should not appear in the formal citation.

5. Line 227-230: The author states a numerical fitting method was used to calculate f(ISOPO2-ISOPOOH). Which numerical fitting method was selected and why? How good is the fitting result? Please include some details of this fitting in the manuscript.

7. What is the physical meaning or the rationale of constructing Equation 5c in the current format? Please note that this is a rather complicated fitting equation and I wonder how did the author obtain such format? Why cannot the equation be expressed in other ways? Could the author include the rationale behind constructing this equation in the text, please?

8. Similar question as the previous one: what are the physical meanings of F? Why is it constructed in such a way? Maybe the author can explain more in the manuscript.

9. In Figure 2, the e-fold formation timescale from the analytical model was plotted against the full box model. However, unlike the analytical model that has a definition and equation for the formation timescale, there is no definition of the formation timescale in the full box model. How did the author calculate the formation timescale in the full box model? Please specify.

10. It seems that PAR3 can capture most of the isoprene-derived SOA values when compared with the full chemistry model. However, at isoprene rich areas such as Amazon forest, central Africa and Southeast Asia, the isoprene-derived SOA was overpredicted by PAR3. Are there any reactions causing such overprediction in isoprene-rich areas? It seems to be some sort of systematical error. The author should discuss the drawbacks of PAR3 (such as overprediction in isoprene-rich areas) so as to objectively evaluate the model and educate potential users.

11. Why would the IEPOX uptake rate peak at around 6 pm? It seems that the only variable in the IEPOX uptake rate (Equation 7) is [OH]*k17, which is affected by temperature and the OH contraction. However, the product of these two values should not cause the IEPOX uptake rate to peak at 6 pm. Could the author explain why he IEPOX uptake rate peak at around 6 pm?

Minor Comments:

Please refrain using the word "it" in Line 269 as it makes the sentence confusing. I suggest the author change this part of the sentence to "because of Equation 9 assumes rapid formation of the SOA."

The author listed the scatter and correlation plots of PAR1 and PAR3 versus the full mechanism box model in Figure S3 and Figure 2. For consistency I suggest include a similar plot for PAR2 in the SI as well.

The phrase "IEPOX condensation rates" in line 391-line 395 should be changed to "IEPOX reactive uptake rates".
* * *

---

## Short Comment (SC1) · 6 Mar 2019

Dear authors,

In my role as Executive editor of GMD, I would like to bring to your attention our Editorial version 1.1:

http://www.geosci-model-dev.net/8/3487/2015/gmd-8-3487-2015.html

This highlights some requirements of papers published in GMD, which is also available

on the GMD website in the 'Manuscript Types' section:

http://www.geoscientific-model-development.net/submission/manuscript_types.html

In particular, please note that for your paper, the following requirements have not been met in the Discussions paper:

- "The main paper must give the model name and version number (or other unique identifier) in the title."

- "If the model development relates to a single model then the model name and the version number must be included in the title of the paper. If the main intention of an article is to make a general (i.e. model independent) statement about the usefulness of a new development, but the usefulness is shown with the help of one specific model, the model name and version number must be stated in the title. The title could have a form such as, "Title outlining amazing generic advance: a case study with Model XXX (version Y)"."

- "All papers must include a section, at the end of the paper, entitled 'Code availability'. Here, either instructions for obtaining the code, or the reasons why the code is not available should be clearly stated. It is preferred for the code to be uploaded as a supplement or to be made available at a data repository with an associated DOI (digital object identifier) for the exact model version described in the paper. Alternatively, for established models, there may be an existing means of accessing the code through a particular system. In this case, there must exist a means of permanently accessing the precise model version described in the paper. In some cases, authors may prefer to put models on their own website, or to act as a point of contact for obtaining the code. Given the impermanence of websites and email addresses, this is not encouraged, and authors should consider improving the availability with a more permanent arrangement. After the paper is accepted the model archive should be updated to include a link to the

GMD paper."

Consequently,

- add a version number for your new parametrization. Even if this is "only" a parametrisation, it might be changed in the future. In this case the version number is important to identify the exact version of the parametrisation used.

- as the main evaluation results are from GEOS-Chem simulations, add this to the title including the version number identifying the modified GEOS-Chem model version unambiguously.

- note that at least the code developments published in the manuscript need to become public themselves. For your article this means that at least the box model needs to be publicly available under a persistent repository or in the supplement BEFORE final acceptance of the paper. It would be best to also make the full modified version of GEOS-Chem available in a public permanent repository, if this is prohibited by license issues, please state this explicitly and add how to get access to the GEOS-Chem code.

Yours,

Astrid Kerkweg

---

## Referee Comment (RC2) · Anonymous Referee #2 · 7 Mar 2019

**General Comments**

The manuscript presents a simplified parameterization of isoprene-epoxydiols (IEPOX) derived secondary organic aerosol (SOA) that exhibits both computational economy and reproducibility of the explicit or full chemistry (Marais et al., 2016) in simulating IEPOX SOA in global chemistry and climate models. The new parameterization was developed based on an approximate analytical fitting in a box model to reproduce the formation yield and time scale of gas-phase intermediates and SOA from isoprene represented by the full chemistry. Three variations of the simplified parameterization

were discussed and evaluated along with fixed yield and VBS against full chemistry in GEOS-Chem v11-02-rc for IEPOX SOA simulations. Simulations by two out of the three new parameterizations generally captured the tropospheric burden of IEPOX-SOA and spatio-temporal profiles of those simulated by the full chemistry while fixed yields and VBS failed to do so. At the same time, the simplified parameterizations were at least 5 times faster than the full chemistry. The study also highlighted the importance of diurnal variation of chemical/meteorological fields to different parameterizations under comparison in the study. As a result, PAR3, the closest to the full chemistry in structure, resembled the full chemistry the best in terms of the response to the diurnal variation of chemical/meteorological fields.

The manuscript is written well and easy to follow. The new parameterization was concluded to be a good alternative to the full chemical mechanism for accurate and fast simulations of IEPOX-SOA in climate model applications. The method used to develop the parameterizations is very repeatable in simplifying other SOA mechanisms and updatable with continuing advance in isoprene SOA mechanisms. Below are several major and minor comments, which need to be addressed and clarified:

Specific Comments Major 1) Organic coating effects were considered as mentioned in Section 2.2 and results were plotted as Figure S1. First of all, the equations and parameters used in the implementation of organic coating effects were not described. Was the resistor representation of uptake coefficient $\gamma$ by Gaston et al., (2014) used here? What were the values of the parameters used then? Was the dependency on the types of organic matters generated in simulations considered? These need to be clarified. Second, the goal of the paper is to improve computational efficiency while retaining the ability to predict ground level IEPOX-SOA relative to the full chemistry. However, the inclusion of organic coating is intended to be realistic, which seems to be beyond the scope. From my understanding, the coating effect was only implemented in the full chemistry where $\gamma$ was modeled explicitly. In other words, the coating effect is now getting "fitted" into the new parametrization intrinsically as well as the uncertainties going along with it. Please clarify whether this is the case or not. If it is, the uncertainties must be discussed. In addition, my worry is that this would make the future efforts to differentiate the inhibiting effects induced by different SOA types under varying environmental conditions hard to implement without fitting a new set of parameters for each type of organic coating. One should expect that variables like organic types, thickness of the coating, and relative humidity would change the effect of organic coating on reactive uptake.

2) Line134-136: overestimation compared to the measurements? The organic coating effect is strongly influenced by the composition of the coating and ambient relative humidity, changing the diffusion coefficient of IEPOX in the coating layer. Therefore, not just the Henry's law constant and the mass accommodation coefficient but the parameterization for organic coating (e.g., diffusion coefficient of IEPOX in the coating layer) could also affect the apparent uptake rate (or heterogenous reaction rate in some literature). It seems that the coating parameterization implemented in this work was not strong enough to counteract the increase in surface as shown by Figure S1, which contradicts with Zhang et al., 2018. Authors should address this or explain why the result is contradictory to the literature. Again, authors should provide the detailed description of the parameterization of the coating effect, and discuss its limitations and uncertainties. 3) Line 140: I found the statement here problematic. Literature effective Henry's law constant for IEPOX spans three orders of magnitude (Gaston et al., 2014; Nguyen et al., 2014; Pye et al., 2013; Sareen et al., 2017), the effect of which on uptake coefficient might not be trivial as stated here. Pye et al., 2013 tested the sensitivity of predicted IEPOX SOA to the Henry's law constant. With an increase in the Henry's law constant by a factor of 7, the predicted IEPOX SOA increased by a factor of 5. This scalability indicates that future update on the Henry's law constant may require a full re-evaluation of the parameterization. Besides, author should also note that the Henry' law constant for IEPOX dissolution into the organic layer would be different from that into aqueous aerosol. Zhang et al., estimated the Henry's law constant for IEPOX into the alpha-pinene SOA coating to be 1-5x10ˆ6 M/atm by fitting a resistor model using

experimental data (Zhang et al., 2018). Authors should justify why Henry's law constant was not altered to accommodate the implementation of organic coating.

4) Figure 3: PAR2 and PAR3 overpredicted IEPOX SOA in source regions. Is this also a result of differing influence of the diurnal variation profiles of chemical fields compared to the full chemistry? Are there any other reasons? Although this paper focuses on evaluating the new parameterization against the full chemistry, the natural question to ask is does it improve the model performance against measurements? No indication was given in that sense. If the full chemistry model with the coating effect tends to overestimate IEPOX SOA vs. measurements in the source regions, the new parametrizations would worsen the model performance. If the full chemistry underesti-mates IEPOX SOA, the overestimation of PAR2 and PAR3 offsets the underestimation to some extent and may improve the model performance. The phrase "more accurate" appears a few times in the manuscript including in the conclusion. It should be more carefully used otherwise misleading.

Minor 1) Figure 1: Which mechanism/parameterization was used to calculate the yields for each step in this figure? Please clarify.

2) Line 400 Figure 4C: Is there a seasonal pattern if the northern and southern hemi-sphere can be plotted separately?

Technical Corrections: Figure 2(a): Was IEPOX-SOA molar yield or IEPOX yield plotted here? The caption didn't match with axis labels.

References

Gaston, C. J., Riedel, T. P., Zhang, Z., Gold, A., Surratt, J. D., & Thornton, J. A. (2014). Reactive Uptake of an Isoprene-Derived Epoxydiol to Submicron Aerosol Parti-cles. Gas/Particle Partitioning and Secondary Organic Aerosol Yields, 48(19), 11178–11186. http://doi.org/10.1021/es5034266 Marais, E. A., Jacob, D. J., Jimenez, J. L., Campuzano-Jost, P., Day, D. A., Hu, W., et al. (2016). Aqueous-phase mechanism

Interactive
comment

for secondary organic aerosol formation from isoprene: application to the southeast United States and co-benefit of SO2 emission controls. Atmospheric Chemistry and Physics, 16(3), 1603–1618. http://doi.org/10.5194/acp-16-1603-2016 Nguyen, T. B., Coggon, M. M., Bates, K. H., Zhang, X., Schwantes, R. H., Schilling, K. A., et al. (2014). Organic aerosol formation from the reactive uptake of isoprene epoxydiols (IEPOX) onto non-acidified inorganic seeds. Atmospheric Chemistry and Physics, 14(7), 3497–3510. http://doi.org/10.5194/acp-14-3497-2014 Pye, H. O. T., Pinder, R. W., Piletic, I. R., Xie, Y., Capps, S. L., Lin, Y.-H., et al. (2013). Epoxide pathways improve model predictions of isoprene markers and reveal key role of acidity in aerosol formation. Environ. Sci. Technol., 47(19), 11056–11064. http://doi.org/10.1021/es402106h Sareen, N., Waxman, E. M., Turpin, B. J., Volkamer, R., & Carlton, A. G. (2017). Potential of Aerosol Liquid Water to Facilitate Organic Aerosol Formation: Assessing Knowledge Gaps about Precursors and Partitioning. Environ. Sci. Technol., 51(6), 3327–3335. http://doi.org/10.1021/acs.est.6b04540

---

## Author Comment (AC1) · 29 May 2019

**Response to reviewers for the paper "A simplified parameterization of isoprene-epoxydiol-derived secondary organic aerosol (IEPOX-SOA) for global chemistry and climate models"**

We thank the reviewers and Executive Editor for their comments on our paper. To guide the review process we have copied the reviewer comments in black text. Our responses are in regular blue font. We have responded to all the referee comments and made alterations to our paper (**in bold text**). For duplicated responses, we referred to prior comments with underlined links and backlinks, to avoid repeating the text.

**Executive editor**

E1.0) In my role as Executive editor of GMD, I would like to bring to your attention our Editorial version 1.1:

http://www.geosci-model-dev.net/8/3487/2015/gmd-8-3487-2015.html

This highlights some requirements of papers published in GMD, which is also available on the GMD website in the 'Manuscript Types' section:

http://www.geoscientific-model-development.net/submission/manuscript_types.html

In particular, please note that for your paper, the following requirements have not been met in the Discussions paper:

- "The main paper must give the model name and version number (or other unique identifier) in the title."

- "If the model development relates to a single model then the model name and the version number must be included in the title of the paper. If the main intention of an article is to make a general (i.e. model independent) statement about the usefulness of a new development, but
the usefulness is shown with the help of one specific model, the model name and version number must be stated in the title. The title could have a form such as, "Title outlining amazing generic advance: a case study with Model XXX (version Y)"."

- "All papers must include a section, at the end of the paper, entitled 'Code availability'. Here, either instructions for obtaining the code, or the reasons why the code is not available should
be clearly stated. It is preferred for the code to be uploaded as a supplement or to be made available at a data repository with an associated DOI (digital object identifier) for the exact model version described in the paper. Alternatively, for established models, there may be an existing means of accessing the code through a particular system. In this case, there must exist a means of permanently accessing the precise model version described in the paper. In
some cases, authors may prefer to put models on their own website, or to act as a point of contact for obtaining the code. Given the impermanence of websites and email addresses, this is not encouraged, and authors should consider improving the availability with a more permanent arrangement. After the paper is accepted the model archive should be updated to include a link to the GMD paper."

Consequently,

- E1.1) add a version number for your new parametrization. Even if this is "only" a parametrisation, it might be changed in the future. In this case the version number is important to identify the exact version of the parametrisation used.

We added a version number to the parameterization (version 1.0), and mentioned this in abstract.

**"Here we present three simplified parameterizations (version 1.0) for IEPOX-SOA simulation, based on an approximate analytical/fitting solution of the IEPOX-SOA yield and formation timescale."**

- E1.2) as the main evaluation results are from GEOS-Chem simulations, add this to the title including the version number identifying the modified GEOS-Chem model version unambiguously.

We added a mention of GEOS-Chem and its version number to the title as follows:

**"A simplified parameterization of isoprene-epoxydiol-derived secondary organic aerosol (IEPOX-SOA) for global chemistry and climate models: a case study with GEOS-Chem v11-02-rc"**

- E1.3) note that at least the code developments published in the manuscript need to become public themselves. For your article this means that at least the box model needs to be publicly available under a persistent repository or in the supplement BEFORE final acceptance of the paper. It would be best to also make the full modified version of GEOS-Chem available in a public permanent repository, if this is prohibited by license issues, please state this explicitly and add how to get access to the GEOS-Chem code.

We recently made the KinSim box model available on the web, as documented in Peng and Jimenez (2019). We have added a citation to this paper.

We have uploaded the different KinSim chemical mechanisms as supplements of the paper, and also to the KinSim chemical mechanisms page.

GEOS-Chem is publicly available in GEOS-Chem wiki site, but we are making the full modified version of GEOS-Chem available upon request, because there are technical details that users should be aware of (e.g. the code also requires corresponding run directory, new restart file is needed, etc.). We updated the Code and Data availability section as follows.

**"Code and Data Availability. The KinSim box model can be downloaded from http://tinyurl.com/kinsim-release (preferred, due to updates) or from the supporting information (https://pubs.acs.org/doi/suppl/10.1021/acs.jchemed.9b00033/suppl_file/ed9b00033_si _001.zip) of Peng and Jimenez (2019). The different KinSim chemical mechanisms used for the box model are available in the supplement of this paper, and also at https://tinyurl.com/kinsim-cases. They can be directly loaded into KinSim to reproduce the calculations in this work by using "Load_Case" function (see detailed usage on http://tinyurl.com/kinsim-help). GEOS-Chem v11-02-rc and meteorological data can be downloaded from GEOS-Chem wiki (http://wiki.seas.harvard.edu/geos-chem/index.php/Downloading_GEOS-Chem_source_code_and_data). GEOS-Chem code modifications for new parameterizations and global model data are available upon email request (duseong.jo@colorado.edu)."**

**Anonymous Referee #1**

R1.0) This manuscript proposed three parameterization methods to reduce the computational cost of simulating Isoprene-derived SOA in GEOS-Chem using the full chemistry model. The
parameterization methods were also compared with the volatility-basis-set and fixed yield methods. The results show that the parameterization methods, especially the third one (PAR3), could generally predict the isoprene-derived SOA spatially while reducing the computational cost of using a full chemistry model. The manuscript also suggests that the VBS approach is not accurate in predicting isoprene-derived SOA because the reactive uptake process is the main
process for isoprene-derived SOA formation.

The manuscript has clear logic and it provides a useful and efficient parameterization method to calculate isoprene-derived SOA in global models such as GEOS-Chem. The quality of the manuscript is good and the main argument is valid, thus the manuscript is worth being published in Geoscientific Model Development. My main comments are that it occurred in a few
places of the manuscript that the assumptions of the model or the detailed processes are not fully clear. I suggest the authors make appropriate changes to the manuscript in the following sections.

R1.1) Line 80: the author states that Pye et al. 2010 used VBS method to predict isoprene derived SOA from first-generation products through partitioning. However, in Pye et al. 2013,
the reactive uptake aqueous pathway was already incorporated to the VBS method in predicting the isoprene-derived SOA. The new VBS results have been improved and are consistent with field measurements. The author should at least incorporate the latest model improvement approach by Pye et al. in the manuscript when discussing the VBS method. With the latest method incorporating 2-methyltetrol and 2-methylglyceric acid, it is inaccurate to state that "the
default VBS mechanism underestimated the observed isoprene SOA formation by a factor of 3 over the southeast US in summer".

Pye et al. (2013) updated the CMAQ model to simulate IEPOX reactive uptake in addition to the Odum 2-product SOA model used as default in CMAQ. This work is analogous to the work by Marais et al. (2016), who added the explicit IEPOX reactive uptake pathway in addition to the
VBS in GEOS-Chem model. We think these works are unrelated with improving the 2-product or VBS scheme, because they added new gas-phase chemistry to simulate IEPOX and SOA formation and particle-phase reaction processes are not simulated within 2-product or VBS framework.

We understand that the current VBS scheme in GEOS-Chem may not include all the recent
findings on SOA formation – wall-corrected yields, volatility-dependent Henry's law constant, photolysis removal, etc. However, the main purpose of this paper is developing a new SOA parameterization and comparing it with current default parameterization options in GEOS-Chem. We have added the following text to Sect. 2.3 to avoid potential reader confusion as follows.

**"We note that there are multiple VBS schemes available in the literature, and their details can vary (e.g., the number of bins, yields, chemical aging, $NO_x$ dependence, photolysis, etc.). In this study we focused on evaluating the current default isoprene VBS scheme in GEOS-Chem."**

R1.2) Line 130-135. The author states that the net effect of the coating effect is to increase the
reaction rate of IEPOX with organic coated aerosols. However, both Gaston et al. 2015 and Zhang et al. 2018 show that the uptake coefficient is highly dependent on the diffusion coefficient of IEPOX in the organic layer, the Herny's law of the IEPOX into the organic layer, etc. With different parameters, the resistor model can give drastically different results. The author should explicitly specify what the equations and values were used to calculate the uptake coefficient in Figure S1. Also, why would the uptake coefficient increase at higher organic loading in Figure S1? As shown in Anttila et al., 2006, Gaston et al. 2015, and Zhang et al. 2018, the resistor model will generate monotonically decreasing reactive uptake values as the coating thickness increases.

We agree that the resistor model results can be substantially changed with different numerical values of the parameters. We added a supplementary section to provide the detailed information for the resistor model equation and parameters (Section 1 in the supplementary information).

**"1 IEPOX reactive uptake coefficient calculation**

**We use the resistor model equation by Gaston et al. (2014b) to calculate the reactive uptake coefficient of IEPOX (γ). The equation is as follows:**

$$\frac{1}{\gamma} = \frac{\omega R_p}{4 D_{gas}} + \frac{1}{\alpha} + \frac{\omega R_p}{4 R T H_{org} D_{org} (q_{org} F - 1)} \tag{S1a}$$

$$F = \frac{\coth(q_{org}) + h(q_{aq}, q_{org}^*)}{1 + \coth(q_{org}) h(q_{aq}, q_{org}^*)} \tag{S1b}$$

$$h(q_{aq}, q_{org}^*) = -\tanh(q_{org}^*) \frac{\frac{H_{aq} D_{aq}}{H_{org} D_{org}} (q_{aq} \coth(q_{aq}) - 1) - (q_{org}^* \coth(q_{org}^*) - 1)}{\frac{H_{aq} D_{aq}}{H_{org} D_{org}} (q_{aq} \coth(q_{aq}) - 1) - (q_{org}^* \tanh(q_{org}^*) - 1)} \tag{S1c}$$

$$q_{org} = R_p \sqrt{\frac{k_{org}}{D_{org}}}, \quad q_{aq} = R_c \sqrt{\frac{k_{aq}}{D_{aq}}}, \quad q_{org}^* = \frac{R_c}{R_p} q_{org} \tag{S1d}$$

**where $\omega$ is the mean molecular speed of IEPOX (m s$^{-1}$), $R_p$ is the particle radius (m), $D_{gas}$ is the gas-phase diffusion coefficient of IEPOX ($10^{-5}$ m$^2$ s$^{-1}$), α is the mass accommodation coefficient (0.1), $R$ is the universal gas constant (8.2057 x $10^{-2}$ L atm mol$^{-1}$ K$^{-1}$), T is temperature (K), $H_{aq}$ and $H_{org}$ are Henry's law coefficients in the aqueous core (1.7 x $10^7$ M atm$^{-1}$) and in the organic layer (2 x $10^6$ M atm$^{-1}$), $D_{aq}$ and $D_{org}$ are diffusion coefficients of IEPOX in the aqueous core ($10^{-9}$ m$^2$ s$^{-1}$) and in the organic layer (discussed below), and $R_c$ is the inorganic aqueous core radius (m). $k_{aq}$ is the first-order reaction rate constant in the aqueous phase (s$^{-1}$), calculated as follows:**

$$k_{aq} = (k_{H^+}[H^+]) + (k_{nuc}[nuc] a_{H^+}) + k_{ga}[ga] \tag{S2}$$

**where $k_{H^+}$ is the reaction rate constant due to acid-catalyzed ring-opening (0.036 M$^{-1}$ s$^{-1}$), [H$^+$] is the proton concentration (M), $a_{H^+}$ is the proton activity, k$_{nuc}$ is the reaction rate constant due to the presence of specific nucleophiles (sulfate and nitrate) (2 x $10^{-4}$ M$^{-1}$ s$^{-1}$), [nuc] is the concentration of nucleophiles (M), k$_{ga}$ is the reaction rate constant due to the presence of general acids (bisulfate) (7.3 x $10^{-4}$ M$^{-1}$ s$^{-1}$), and [ga] is the concentration of**

**general acids (M). We assumed the reaction rate coefficient of IEPOX in the organic layer ($k_{org}$) is the same as $k_{aq}$. We note that the equation above is different from the IEPOX reactive uptake equation used by Zhang et al. (2018), which is based on Gaston et al. (2014a). The equation from Gaston et al. (2014a) can be derived from the Taylor series approximation by assuming thin coatings (Anttila et al., 2006). Therefore, we used the**
**equation S1 to avoid some possible errors from the cases that second or higher order Taylor terms become important.**

**The diffusion coefficient of IEPOX in the organic layer ($D_{org}$) substantially changes by several orders of magnitude over a range of relative humidity (RH) in the atmosphere. Based on Table S3 of Zhang et al. (2018), we considered the RH dependence for $D_{org}$ values.**
**Table S1 show $D_{org}$ values we used for GEOS-Chem calculation.**

[Figure]

**Figure S1. The diffusion coefficient of IEPOX in the organic layer ($D_{org}$) as a function of RH. Red points indicate values calculated by Zhang et al. (2018). Values in between red points are log-linearly interpolated, and values below 30% RH or above 90% are set to be the**
**constant values."**

For the uptake coefficient at higher organic loadings, we found an error in our calculation code. It was OK for lower organic loadings, but the bug erroneously increased the uptake coefficient by 30% at higher organic loading conditions. We corrected this error for all the new GEOS-Chem runs shown in the revised paper. New results are shown in Figs. S2 and S3 with the updated main
text.

**"Standard GEOS-Chem assumes no organic coating; only the surface area of inorganic aerosols. We updated the model to include suppression of IEPOX reactive uptake by the**

organic coating, and to use the available surface area of the total sulfate-ammonium-nitrate-organic aerosols mixture at a given relative humidity with hygroscopic growth factors. We found that the IEPOX reactive uptake coefficient ($\gamma$) was always decreased at atmospheric relevant aerosol pH and relative humidity conditions, but the IEPOX reactive uptake rate constant increased in some conditions (high pH and high IEPOX diffusion coefficient in the organic layer, Fig. S2). We note that this is the case for GEOS-Chem v11-02-rc, because GEOS-Chem does not take into account organic aerosol mass for aerosol radius and aerosol surface area calculation when it calculates IEPOX reactive uptake. Therefore, additional OA mass considered in this study increases available aerosol surface area for IEPOX reactive uptake, which compensates or sometimes overcomes the effects by the decrease of $\gamma$ as shown in Eq. (1) for the first-order uptake rate constant of IEPOX to form IEPOX-SOA:

$$\text{IEPOX uptake rate constant} = \frac{S_a}{\dfrac{r_a}{D_g} + \dfrac{4}{\gamma \times v_{mms}}} \tag{1}$$

$S_a$ is the wet aerosol surface area on which IEPOX can be taken up ($m^2\ m^{-3}$), $r_a$ is the wet aerosol radius (m), $D_g$ is gas-phase diffusion coefficient of IEPOX ($m^2\ s^{-1}$), and $v_{mms}$ is the mean molecular speed ($m\ s^{-1}$) of gas-phase IEPOX. Again, the effects of organic coating on IEPOX uptake rate constant in this study can be different from previous observational studies (Hu et al., 2016; Zhang et al., 2018), because observational studies used the measured and fixed available aerosol surface area and radius, and they changed organic aerosol layer thickness for their calculations (i.e. inorganic core radius was changed but total particle radius and surface area were not changed). When we assumed the fixed aerosol radius and aerosol surface area, and only organic coating thickness increased as OA mass increased, all the case showed the decreasing IEPOX reactive uptake rate constants (Fig. S3)."

[Figure]

Figure S2. IEPOX reactive uptake coefficient (a,b,c) and uptake rate constant (d,e,f) as a function of OA mass concentrations. Different colors indicate $D_{org}$ values ranging from 6 x $10^{-18}$ to 1 x $10^{-12}$ $m^{-2}\ s^{-1}$. Aerosol pH values were set to be -1 (a,d), 1 (b,e), and 3 (c,f),

**respectively. Sulfate aerosol mass concentration was assumed as 10 μg m$^{-3}$. Densities of sulfate and organic aerosols were set to be 1.7 and 1.3 g cm$^{-3}$, respectively, based on densities used by GEOS-Chem v11-02-rc. Initial aerosol radius of 50 nm and aerosol surface area of 3 x 10$^{-6}$ cm$^2$ cm$^{-3}$ were assumed for organic aerosol mass = 0 μg m$^{-3}$. The**
**changes of aerosol radius and aerosol surface area were calculated as OA mass increases, and aerosol inorganic core radius was fixed as 50 nm.**

[Figure]

**Figure S3. IEPOX reactive uptake coefficient (a,b,c) and uptake rate constant (d,e,f) as a function of OA mass concentrations. Different colors indicate D$_{org}$ values ranging from 6 x**
**10$^{-18}$ to 1 x 10$^{-12}$ m$^{-2}$ s$^{-1}$. Aerosol pH values were set to be -1 (a,d), 1 (b,e), and 3 (c,f), respectively. Sulfate aerosol mass concentration was assumed as 1 μg m$^{-3}$. Densities of sulfate and organic aerosols were set to be 1.7 and 1.3 g cm$^{-3}$, respectively, based on densities used by GEOS-Chem v11-02-rc. Initial aerosol radius of 50 nm and aerosol surface area of 3 x 10$^{-6}$ cm$^2$ cm$^{-3}$ were assumed for organic aerosol mass = 0 μg m$^{-3}$. The**
**changes of aerosol radius and aerosol surface area were fixed regardless of OA mass increase. Aerosol core radius was reduced in proportion to the OA mass increase (i.e. coating thickness increase).**

We re-ran GEOS-Chem model to include effects of RH on D$_{org}$. In addition, soil NO$_x$ emissions were erroneously set as zero for the runs shown in the GMDD paper. We also corrected this
error. As a result, Figures 1, 3, 4, 5, and 6 were updated, but conclusions remained the same (PAR2 and PAR3 again showed the best performance). We have updated all the figures and numbers in the main text.

(Backlinks to R1.10, R2.1 and R2.2)

R1.3) Line 140. I understand that the author is intended to cross compare parameterizations
against the full chemistry model for isoprene formation, however it is inaccurate and a bit misleading to state that "Updating the parameterizations developed here with more accurate values of H or determined in future literature studies would be trivial." The H or values affect the absolute values of the isoprene-derived SOA significantly, and currently the estimation of the Herny's law constant can vary by two orders of magnitudes. Knowing an accurate value of the
Herny's law constant will help bridge the gap between the model and the field measurement, thus these values are not trivial. I suggest the author revise this sentence.

We wanted to deliver the message that our parameterizations do not require any complex refitting processes even if parameters are updated from the future study. I.e. *updating the parameterization with more accurate literature values of H* is trivial, even if *determining accurate values of H* is far from trivial. To avoid possibly misleading the readers with this sentence, we have added the following text:

**"Parameters used in this study such as the Henry's law constant and the IEPOX diffusion coefficient in OA can be easily updated in future studies. Our parameterizations are flexible to the change of these variables, because they use the IEPOX reactive uptake rate ($k_{18}$ in Eqs. (7) and (14) in Sect. 3) rather than using individual input parameters. Therefore, updating the parameterizations developed here with more accurate values of input parameters determined in future literature studies is easy without having to refit the parameterizations."**

(Backlink to R2.3)

R1.4) Please remove the Nault et al. in prep. citation as unpublished work should not appear in the formal citation.

We changed this citation to that of the the AGU conference abstract:

**"Effects of sea salt on pH and detailed analysis against the aircraft measurements were discussed in detail by Nault et al. (2018)."**

**Nault, B. A., Campuzano-Jost, P., Douglas Day, Hu, W., Palm, B., Schroder, J. C., Bahreini, R., Bian, H., Chin, M., Clegg, S. L., Colarco, P. R., Crounse, J. D., Dibb, J. E., Kim, M. J., Kodros, J., Lopez-Hilfiker, F., Marais, E. A., Middlebrook, A. M., Neuman, J. A., Nowak, J. B., Pierce, J. R., Scheuer, E. M., Thornton, J. A., Veres, P. R., Wennberg, P. O. and Jimenez, J. L.: Global Survey of Submicron Aerosol Acidity (pH), Abstract A53A-06 presented at American Geophysical Union Fall Meeting 2018, 10-14, December, Washington, D.C., 2018.**

R1.5) Line 227-230: The author states a numerical fitting method was used to calculate f(ISOPO2-ISOPOOH). Which numerical fitting method was selected and why? How good is the fitting result? Please include some details of this fitting in the manuscript.

We attempted various functional forms, independent variables, and initial guesses to get the best results compared to the box model results. We added the following explanation to the main text:

**"We tried different functional forms for the equation (polynomial, Gaussian, Lorentzian, exponential, double-exponential, trigonometric, Hill, Sigmoid, etc.), independent variables, and initial guesses for the coefficients. We found that the Hill type equation combined with the production term of $ISOPO_2$ in exponential form showed the best results compared to the box model calculation."**

The fitting results were very accurate as shown in Figs. 2a and S5a. Because the other parts can be analytically calculated for IEPOX-SOA yield, the fitting for $ISOPO_2$ self-reaction is the only factor causing the uncertainty of our parameterizations.

(Backlinks to R1.6 and R1.7)

R1.6) What is the physical meaning or the rationale of constructing Equation 5c in the current format? Please note that this is a rather complicated fitting equation and I wonder how did the author obtain such format? Why cannot the equation be expressed in other ways? Could the author include the rationale behind constructing this equation in the text, please?

See response to comment R1.5.

R1.7) Similar question as the previous one: what are the physical meanings of F? Why is it constructed in such a way? Maybe the author can explain more in the manuscript.

We added the following text to explain the physical meaning of F. The functional form was derived from trial and error as per R1.5.

**"We designed the term F to consider contributions of high and low $NO_x$ pathways to the formation timescale in the single equation system. $ISOPO_2$ + NO pathway is dominant when F = 0 and $ISOPO_2$ + $HO_2$ pathway is dominant when F = 1. F cannot be below 0 or above 1 in terms of the physical meaning, but the fitted F can have values outside of 0 to 1 because the numerical fitting works to minimize the total error compared to the box model calculated timescale of IEPOX-SOA"**

R1.8) In Figure 2, the e-fold formation timescale from the analytical model was plotted against the full box model. However, unlike the analytical model that has a definition and equation for the formation timescale, there is no definition of the formation timescale in the full box model. How did the author calculate the formation timescale in the full box model? Please specify.

We added the text describing the detailed calculation process of the formation timescale from the box model to the caption of Fig. 2:

**"Formation timescale of the full mechanism box model was calculated as follows. We saved IEPOX concentrations for each timestep. We defined the formation timescale as the time when the IEPOX concentration is closest to the 1 - 1/e (~63%) of the final IEPOX concentration."**

R1.9) It seems that PAR3 can capture most of the isoprene-derived SOA values when compared with the full chemistry model. However, at isoprene rich areas such as Amazon forest, central Africa and Southeast Asia, the isoprene-derived SOA was overpredicted by PAR3. Are there any reactions causing such overprediction in isoprene-rich areas? It seems to be some sort of systematical error. The author should discuss the drawbacks of PAR3 (such as overprediction in isoprene-rich areas) so as to objectively evaluate the model and educate potential users.

We figured out that our parameterizations generally overestimate the IEPOX-SOA when OH concentrations are low (newly added Fig. S9), and OH concentrations are low in Amazon, central Africa, and Southeast Asia regions (Fig. S10). We added the following text with two additional supplemental figures (Figs. S9 and S10) to document these points:

**"Contrary to PAR1, which calculated IEPOX-SOA yield at the time of isoprene emission, PAR2 and PAR3 did not show a global underestimation because they only calculated IEPOX yield at the time of isoprene emission, and then simulated the IEPOX reactive uptake explicitly. However, they showed slight overestimations over isoprene source regions such as the Amazon. We found that PAR2 and PAR3 generally overestimate the IEPOX-**

SOA when OH concentrations are low (Fig. S9), and the Amazon is one of low OH regions from GEOS-Chem model (Fig. S10). We attributed this tendency to the effects of lifetime of IEPOX precursor gases, for which OH concentrations are one of the major controlling factors. IEPOX yields in PAR2 and PAR3 are calculated using the instantaneous chemical fields. Therefore, the discrepancies between the explicit chemistry and PAR2-3 are reduced when the lifetimes of precursor gases are short. For the southeastern US where PAR3 did not show an overestimation, the lifetimes of isoprene and ISOPOOH were 0.9 hours and 1.5 hours, respectively. The discrepancies are much larger for the Amazon, the lifetimes of isoprene and ISOPOOH are 12.3 hours and 6.1 hours, respectively, due to low OH concentrations. As a result, the PAR1-3 calculated the similar IEPOX production rate (1.9 Tg yr$^{-1}$) from the ISOPOOH + OH reaction compared to the full chemistry (1.8 Tg yr$^{-1}$) for the southeastern US, but the disagreement was larger for the Amazon (4.8 Tg yr$^{-1}$ in the PAR1-3 vs 3.9 Tg yr$^{-1}$ in the full chemistry). We anticipate that the discrepancy in source regions will be reduced in the future version of GEOS-Chem, because GEOS-Chem with the most up to date isoprene mechanism predicts higher OH concentrations (up to 250% increase) in Amazon, central Africa, and Borneo regions compared to the isoprene mechanism used in this study (Fig. S17 in Bates and Jacob, 2019)."

[Figure]

**Figure S9. Scatterplots of the IEPOX-SOA concentration ratio (five parameterizations against the explicit full chemistry) vs. OH concentration within the PBL. Each point represents the monthly averaged model grid value for four major isoprene source regions [the Southeastern United States: 30°N – 40°N, 100°W – 80°W, Amazon: 10°S – 0°S, 70°W –**
**60°W, Central Africa: 5°N – 15°N, 10°E – 30°E, Borneo: 5°S – 5°N, 105°E – 120°E]. Colors indicate the IEPOX-SOA concentration simulated by the full chemistry.**

[Figure]

**Figure S10. Global annual mean OH concentrations for July 2013 – June 2014 as predicted by the GEOS-Chem v11-02-rc used in this study.**

(Backlink to R2.4)

R1.10) Why would the IEPOX uptake rate peak at around 6 pm? It seems that the only variable in the IEPOX uptake rate (Equation 7) is [OH]*k17, which is affected by temperature and the OH contraction. However, the product of these two values should not cause the IEPOX uptake rate to peak at 6 pm. Could the author explain why he IEPOX uptake rate peak at around 6 pm?

First of all, the IEPOX uptake rate peak changed after we re-ran the model with the updated diffusion coefficient of IEPOX in OA, which changed the organic coating effect (see response to comment R1.2). IEPOX uptake rate peaks at around 4 p.m., which is used by both the full chemistry and parameterizations. There are many variables affecting the IEPOX uptake rate constant ($k_{18}$) – aerosol pH, aerosol surface area, temperature, organic coating thickness, etc. [OH]*$k_{17}$ does not direcly affect the IEPOX uptake rate, although it is an important reaction impacting the IEPOX concentration. We found that the diurnal variation of the IEPOX uptake rate constant generally followed aerosol pH variations, which in turn was affected by aerosol water due to relative humidity changes. Aerosol pH was low around at 4 p.m. because $H^+$ aerosol concentration increased due to low aerosol water content. $R^2$ between aerosol pH and aerosol water was 0.85. We added the diurnal variation of aerosol pH to Fig. S8.

[Figure]

**Figure S8. Diurnal variations of chemical/meteorological fields used in box model calculation (Fig. 5d). Values were extracted from GEOS-Chem global mean results for four major isoprene source regions [the Southeastern United States: 30°N – 40°N, 100°W – 80°W, Amazon: 10°S – 0°S, 70°W – 60°W, Central Africa: 5°N – 15°N, 10°E – 30°E, Borneo: 5°S – 5°N, 105°E – 120°E]. Figures represent approximate annual mean diurnal variation profiles, which were calculated by using the first two days of each month model outputs at 30 minutes interval averaged within the PBL, averaging points of the same local time.**

Minor Comments:

R1.11) Please refrain using the word "it" in Line 269 as it makes the sentence confusing. I suggest the author change this part of the sentence to "because of Equation 9 assumes rapid formation of the SOA."

We removed "it" and rephrase the sentence as follows:

**"As a result, if the yield from Eq. (9) is used for making IEPOX-SOA, chemical transport models would likely overestimate IEPOX-SOA concentrations locally in isoprene-emitting areas due to the instantaneous formation of IEPOX-SOA from Eq. (9)."**

R1.12) The author listed the scatter and correlation plots of PAR1 and PAR3 versus the full
mechanism box model in Figure S3 and Figure 2. For consistency I suggest include a similar plot for PAR2 in the SI as well.

PAR2 and PAR3 have the same yield of IEPOX from isoprene ($Y_{IEPOX}$). PAR3 was developed on top of PAR2 by introducing an intermediate species ($SOAP_I$). Therefore, Figure 2a covers both PAR2 and PAR3 (Figure 2b is only for PAR3 since PAR2 is not using the formation timescale).
We clarified this in the caption of Fig. 2 as follows:

**"Scatterplots of the results of parameterizations (y-axis) versus the full mechanism (x-axis) box model results for (a) IEPOX molar yield (PAR2 and PAR3) and (b) formation timescale (PAR3)."**

R1.13) The phrase "IEPOX condensation rates" in line 391-line 395 should be changed to "IEPOX reactive uptake rates".

We have updated this text as follows:
**"For example, the instantaneous IEPOX-SOA yield using both isoprene emission and IEPOX reactive uptake rate constant at noon is lower than the yield calculated using the**
**isoprene emission rate at 12 p.m. and the IEPOX reactive uptake rate constant at 6 p.m, when each process peaks, explaining most of the underestimation in PAR1. Contrary to PAR1, which calculated IEPOX-SOA yield at the time of isoprene emission, PAR2 and PAR3 did not show a global underestimation because they only calculated IEPOX yield at the time of isoprene emission, and then simulated the IEPOX reactive uptake explicitly."**

**Anonymous Referee #2**

R2.0) General Comments

The manuscript presents a simplified parameterization of isoprene-epoxydiols (IEPOX) derived secondary organic aerosol (SOA) that exhibits both computational economy and reproducibility of the explicit or full chemistry (Marais et al., 2016) in simulating IEPOX SOA in global chemistry and climate models. The new parameterization was developed based on an approximate analytical fitting in a box model to reproduce the formation yield and time scale of gas-phase intermediates and SOA from isoprene represented by the full chemistry. Three variations of the simplified parameterization were discussed and evaluated along with fixed yield and VBS against full chemistry in GEOS-Chem v11-02-rc for IEPOX SOA simulations. Simulations by two out of the three new parameterizations generally captured the tropospheric burden of IEPOX-SOA and spatio-temporal profiles of those simulated by the full chemistry while fixed yields and

VBS failed to do so. At the same time, the simplified parameterizations were at least 5 times faster than the full chemistry. The study also highlighted the importance of diurnal variation of chemical/meteorological fields to different parameterizations under comparison in the study. As a result, PAR3, the closest to the full chemistry in structure, resembled the full chemistry the best in terms of the response to the diurnal variation of chemical/meteorological fields.

The manuscript is written well and easy to follow. The new parameterization was concluded to be a good alternative to the full chemical mechanism for accurate and fast simulations of IEPOX-SOA in climate model applications. The method used to develop the parameterizations is very repeatable in simplifying other SOA mechanisms and updatable with continuing advance in isoprene SOA mechanisms. Below are several major and minor comments, which need to be addressed and clarified:

Specific Comments Major

R2.1) Organic coating effects were considered as mentioned in Section 2.2 and results were plotted as Figure S1. First of all, the equations and parameters used in the implementation of organic coating effects were not described. Was the resistor representation of uptake coefficient by Gaston et al., (2014) used here? What were the values of the parameters used then? Was the dependency on the types of organic matters generated in simulations considered? These need to be clarified. Second, the goal of the paper is to improve computational efficiency while retaining the ability to predict ground level IEPOX-SOA relative to the full chemistry. However, the inclusion of organic coating is intended to be realistic, which seems to be beyond the scope.

From my understanding, the coating effect was only implemented in the full chemistry where was modeled explicitly. In other words, the coating effect is now getting "fitted" into the new parametrization intrinsically as well as the uncertainties going along with it. Please clarify whether this is the case or not. If it is, the uncertainties must be discussed. In addition, my worry is that this would make the future efforts to differentiate the inhibiting effects induced by different

SOA types under varying environmental conditions hard to implement without fitting a new set of parameters for each type of organic coating. One should expect that variables like organic types, thickness of the coating, and relative humidity would change the effect of organic coating on reactive uptake.

We used the resistor model equation by Gaston et al. (2014) in ACP to calculate IEPOX
reactive uptake coefficient. We added a supplementary section to describe the detailed procedure and parameters for the resistor model (See the response to the comment R1.2 for details). We did not take into account the changes of IEPOX diffusivity in the organic layer by different SOA types, although this could be easily done in future work by updating the numerical value of that parameter in different regions / times in the model as needed. The diffusivity as a function of relative humidity was newly considered in the revised text (See the response to the comment R1.2). As a result, Figures 1, 3, 4, and 5 were changed, but conclusions remained the same (PAR2 and PAR3 again showed the best performance). We updated figures and numbers in the main text.

The coating effect was implemented not only in the full chemistry but in the parameterizations. Our parameterizations explicitly calculate IEPOX reactive uptake coefficient and rate as does the full chemistry, which avoids having to refit the parameterizations when updated individual parameters become available (e.g. Henry's law constant, diffusivity). As a result, future works making IEPOX reactive uptake rate change can be explicitly reflected in our parameterizations. We clarified this in the main text as follows:

**"Parameters used in this study such as the Henry's law constant and IEPOX diffusion coefficient in OA can be updated in future studies, as new information becomes available in the literature. Our parameterizations are flexible to the change of these variables, because they use the IEPOX reactive uptake rate constant ($k_{18}$ in Eqs. (7) and (14) in Sect. 3) rather than using individual input parameters. Therefore, updating the parameterizations developed here with more accurate values of input parameters determined in future literature studies can be done without refitting."**

R2.2) Line134-136: overestimation compared to the measurements? The organic coating effect is strongly influenced by the composition of the coating and ambient relative humidity, changing the diffusion coefficient of IEPOX in the coating layer. Therefore, not just the Henry's law constant and the mass accommodation coefficient but the parameterization for organic coating (e.g., diffusion coefficient of IEPOX in the coating layer) could also affect the apparent uptake rate (or heterogenous reaction rate in some literature). It seems that the coating parameterization implemented in this work was not strong enough to counteract the increase in surface as shown by Figure S1, which contradicts with Zhang et al., 2018. Authors should address this or explain why the result is contradictory to the literature. Again, authors should provide the detailed description of the parameterization of the coating effect, and discuss its limitations and uncertainties.

We examined factors making our calculation deviate from the Figure 1 in Zhang et al. (2018). We found two major differences between our work and Zhang et al.'s work. First one is the diffusivity of IEPOX in the organic layer ($D_{org}$). We used a $D_{org}$ value of $5 \times 10^{-10}$ m$^2$ s$^{-1}$ (Gaston et al., 2014a) in the GMDD version of this paper, while Zhang et al. (2018) used lower $D_{org}$ values ($6 \times 10^{-18}$ and $2.7 \times 10^{-17}$ m$^2$ s$^{-1}$ at 30% and 50% RH, respectively). As shown in Fig. S2, $D_{org}$ value of $5 \times 10^{-10}$ m$^2$ s$^{-1}$ makes the IEPOX reactive uptake rate increase in all conditions (d,e,f in Fig. S2), but the IEPOX reactive uptake rate decreases in acidic conditions with lower $D_{org}$ values.

Second, we calculated the case that additional OA mass coated on the surface of inorganic aerosol, which increased the aerosol radius and aerosol surface area (i.e. $R_c$ was fixed, but $R_p$ and $S_a$ were increased.). We calculated the organic coating effects in this way for GEOS-Chem, because the GEOS-Chem standard version only uses inorganic aerosol radius and surface area to calculate IEPOX reactive uptake. However, Zhang et al. (2018) fixed the aerosol radius and surface area, and examined the effects of organic coating thickness on the IEPOX reactive uptake (i.e. $R_p$ and $S_a$ were fixed, but $R_c$ was decreased as organic coating increased). We clarified this point in the main text, and we added a Fig. S3 to show IEPOX reactive uptake rate change for the case that $R_p$ and $S_a$ were fixed (similar to the case by Zhang et al., 2018).

See our response to the comment R1.2 for details.

R2.3) Line 140: I found the statement here problematic. Literature effective Henry's law constant for IEPOX spans three orders of magnitude (Gaston et al., 2014; Nguyen et al., 2014; Pye et al., 530   2013; Sareen et al., 2017), the effect of which on uptake coefficient might not be trivial as stated here. Pye et al., 2013 tested the sensitivity of predicted IEPOX SOA to the Henry's law constant. With an increase in the Henry's law constant by a factor of 7, the predicted IEPOX SOA increased by a factor of 5. This scalability indicates that future update on the Henry's law constant may require a full re-evaluation of the parameterization. Besides, author should also 535   note that the Henry's law constant for IEPOX dissolution into the organic layer would be different from that into aqueous aerosol. Zhang et al., estimated the Henry's law constant for IEPOX into the alpha-pinene SOA coating to be 1-5x10ˆ6 M/atm by fitting a resistor model using experimental data (Zhang et al., 2018). Authors should justify why Henry's law constant was not altered to accommodate the implementation of organic coating.

See our response to comment R1.3.

We used Henry's law constant in the organic layer as $2 \times 10^6$ M atm$^{-1}$ based on Zhang et al. (2018). We clarified it in the supplement text.

R2.4) Figure 3: PAR2 and PAR3 overpredicted IEPOX SOA in source regions. Is this also a 545   result of differing influence of the diurnal variation profiles of chemical fields compared to the full chemistry? Are there any other reasons? Although this paper focuses on evaluating the new parameterization against the full chemistry, the natural question to ask is does it improve the model performance against measurements? No indication was given in that sense. If the full chemistry model with the coating effect tends to overestimate IEPOX SOA vs. measurements in 550   the source regions, the new parametrizations would worsen the model performance. If the full chemistry underestimates IEPOX SOA, the overestimation of PAR2 and PAR3 offsets the underestimation to some extent and may improve the model performance. The phrase "more accurate" appears a few times in the manuscript including in the conclusion. It should be more carefully used otherwise misleading.

The parameterizations developed in this study were designed to reproduce results by the full explicit chemistry. The full chemistry has been evaluated against measurements in prior publications (e.g. Marais et al., 2016). Currently, there are many uncertain parameters substantially affecting the concentrations of IEPOX-SOA (e.g. Henry's law constant and IEPOX diffusivity in OA). We believe that the explicit full chemistry will be improved with future 560   laboratory, field, and modeling studies, and the parameterizations can be updated easily to capture those advances, as long as the structure of the full chemistry mechanism does not change. Therefore, we think addressing the performance against the measurements is beyond the scope of our study, because it involves consideration from various uncertain parameters.

See also the response to R1.9.

Minor

R2.5) Figure 1: Which mechanism/parameterization was used to calculate the yields for each step in this figure? Please clarify.

We clarified it in the caption as follows. We changed Figure 1 because we re-ran the model to use RH-dependent $D_{org}$ values from Zhang et al. (2018).

**"Figure 1. Schematic diagrams of IEPOX-SOA chemistry for (a) $HO_2$ and (b) NO dominant regions. Blue arrows indicate IEPOX-SOA formation pathways and red arrows represent other chemical pathways that do not form significant IEPOX-SOA. Values are averaged molar yields relative to the initial oxidation amount of isoprene from GEOS-Chem v11-02-rc results using the explicit full chemistry with updates in this study (see Sect. 2.2) over Borneo (as an example of $HO_2$ dominant conditions, 5°S – 5°N, 105°E – 120°E) and Beijing (as an example of NO dominant conditions, 35°N – 45°N, 110°E – 120°E) from July 2013 to June 2014. We note that Beijing is located in a region with typically low isoprene emissions, so the appreciable yield of IEPOX-SOA will still result in small ambient concentrations."**

R2.6) Line 400 Figure 4C: Is there a seasonal pattern if the northern and southern hemisphere can be plotted separately?

We added a new figure (Fig. S12) to separate the seasonal patterns of the global burdens in both hemispheres. We confirmed that PAR2 and PAR3 again showed better results. We added the following text accordingly:

**"It also applied to the seasonal patterns of the hemispheric burden when we separated them for the northern and southern hemispheres, as shown in Fig. S12."**

[Figure]

**Figure S12. Same as Fig. 4c but for (a) Northern and (b) Southern Hemisphere.**

R2.7) Technical Corrections: Figure 2(a): Was IEPOX-SOA molar yield or IEPOX yield plotted here? The caption didn't match with axis labels.

We changed the figure caption to correct it.

**References**

Anttila, T., Kiendler-Scharr, A., Tillmann, R. and Mentel, T. F.: On the Reactive Uptake of Gaseous Compounds by Organic-Coated Aqueous Aerosols: Theoretical Analysis and Application to the Heterogeneous Hydrolysis of N2O5, J. Phys. Chem. A, 110, 10435–10443, doi:10.1021/jp062403c, 2006.

Bates, K. H. and Jacob, D. J.: A new model mechanism for atmospheric oxidation of isoprene: global effects on oxidants, nitrogen oxides, organic products, and secondary organic aerosol, Atmos. Chem. Phys. Discuss., 2019, 1–46, doi:10.5194/acp-2019-328, 2019.

Gaston, C. J., Riedel, T. P., Zhang, Z., Gold, A., Surratt, J. D. and Thornton, J. A.: Reactive uptake of an isoprene-derived epoxydiol to submicron aerosol particles, Environ. Sci. Technol., 48, 11178–11186, 2014a.

Gaston, C. J., Thornton, J. A. and Ng, N. L.: Reactive uptake of N2O5 to internally mixed inorganic and organic particles: the role of organic carbon oxidation state and inferred organic phase separations, Atmos. Chem. Phys., 14, 5693–5707, 2014b.

Hu, W., Palm, B. B., Day, D. A., Campuzano-Jost, P., Krechmer, J. E., Peng, Z., De Sa Suzane, S., Martin, S. T., Alexander, M. L., Baumann, K., Hacker, L., Kiendler-Scharr, A., Koss, A. R., De Gouw, J. A., Goldstein, A. H., Seco, R., Sjostedt, S. J., Park, J. H., Guenther, A. B., Kim, S., Canonaco, F., Prévôt, A. S. H., Brune, W. H. and Jimenez, J. L.: Volatility and lifetime against OH heterogeneous reaction of ambient isoprene-epoxydiols-derived secondary organic aerosol (IEPOX-SOA), Atmos. Chem. Phys., 16, 11563–11580, doi:10.5194/acp-16-11563-2016, 2016.

Marais, E. A., Jacob, D. J., Jimenez, J. L., Campuzano-Jost, P., Day, D. A., Hu, W., Krechmer, J., Zhu, L., Kim, P. S., Miller, C. C., Fisher, J. A., Travis, K., Yu, K., Hanisco, T. F., Wolfe, G. M., Arkinson, H. L., Pye, H. O. T., Froyd, K. D., Liao, J. and McNeill, V. F.: Aqueous-phase mechanism for secondary organic aerosol formation from isoprene: Application to the southeast United States and co-benefit of SO2 emission controls, Atmos. Chem. Phys., 16, 1603–1618, doi:10.5194/acp-16-1603-2016, 2016.

Pye, H. O. T., Pinder, R. W., Piletic, I. R., Xie, Y., Capps, S. L., Lin, Y.-H., Surratt, J. D., Zhang, Z., Gold, A. and Luecken, D. J.: Epoxide pathways improve model predictions of isoprene markers and reveal key role of acidity in aerosol formation, Environ. Sci. Technol., 47, 11056–11064, 2013.

Zhang, Y., Chen, Y., Lambe, A. T., Olson, N. E., Lei, Z., Craig, R. L., Zhang, Z., Gold, A., Onasch, T. B., Jayne, J. T., Worsnop, D. R., Gaston, C. J., Thornton, J. A., Vizuete, W., Ault, A. P. and Surratt, J. D.: Effect of the Aerosol-Phase State on Secondary Organic Aerosol Formation from the Reactive Uptake of Isoprene-Derived Epoxydiols (IEPOX), Environ. Sci. Technol. Lett., acs.estlett.8b00044, doi:10.1021/acs.estlett.8b00044, 2018.